# Actin Bundles Dynamics and Architecture

**DOI:** 10.3390/biom13030450

**Published:** 2023-02-28

**Authors:** Sudeepa Rajan, Dmitri S. Kudryashov, Emil Reisler

**Affiliations:** 1Department of Chemistry and Biochemistry, University of California, Los Angeles, CA 90095, USA; 2Department of Chemistry and Biochemistry, Ohio State University, Columbus, OH 43210, USA; 3Molecular Biology Institute, University of California, Los Angeles, CA 90095, USA

**Keywords:** actin bundles, fascin, α-actinin, espin, plastin/fimbrin

## Abstract

Cells use the actin cytoskeleton for many of their functions, including their division, adhesion, mechanosensing, endo- and phagocytosis, migration, and invasion. Actin bundles are the main constituent of actin-rich structures involved in these processes. An ever-increasing number of proteins that crosslink actin into bundles or regulate their morphology is being identified in cells. With recent advances in high-resolution microscopy and imaging techniques, the complex process of bundles formation and the multiple forms of physiological bundles are beginning to be better understood. Here, we review the physiochemical and biological properties of four families of highly conserved and abundant actin-bundling proteins, namely, α-actinin, fimbrin/plastin, fascin, and espin. We describe the similarities and differences between these proteins, their role in the formation of physiological actin bundles, and their properties—both related and unrelated to their bundling abilities. We also review some aspects of the general mechanism of actin bundles formation, which are known from the available information on the activity of the key actin partners involved in this process.

## 1. Introduction

Actin, one of the key and highly conserved elements of the cytoskeleton, amounts to approximately 5–15% of total cell proteins [1,2]. It is indispensable for driving many cellular processes, including cell migration, cytokinesis, vesicle transport, and contractile force generation [3]. The globular actin monomers (G-actin) polymerize to form semi-flexible double-stranded helical filaments (F-actin), also known as microfilaments. To perform diverse cellular functions, these filaments assemble into higher-order structures, such as branched networks or bundles. More than 150 actin-binding proteins (ABPs) are known to associate with the actin cytoskeleton, and many of them regulate actin functions [4]. These proteins are involved in: (1) regulation of actin assembly and disassembly, (2) actin-driven movements in cells, (3) connecting actin structures to plasma membrane/cell organelles or other cytoskeleton proteins, and (4) organizing actin filaments (by their crosslinking) into higher-order structures, such as branched actin networks or actin bundles [1,5,6].

It is now recognized that complex actin networks and bundles are essential for many cellular functions. Accordingly, proteins forming higher-order structures are among the most represented and diverse functional families of actin-binding proteins. Lately, actin-bundling proteins have been attracting a lot of attention as their malfunction is linked to malignant cancers, muscular dystrophy, bone disease, and immunological disorders [7,8,9,10,11].

The main aim of this review is to integrate and summarize current biochemical and structural information on several major actin-bundling proteins, with a primary focus on proteins present at the leading edge of cells and in membrane protrusions. This includes α-actinin, fimbrin/plastin, fascin, and espin. We also briefly discuss the assembly and disassembly mechanisms of parallel actin bundles that are an integral part of microvilli, stereocilia, and filopodia. We hope that this information will assist the interested reader in understanding the physiochemical properties of actin bundles and their structure, assembly, disassembly, and biological functions.

## 2. Actin Organization in the Cell

Actin polymerizes to form double-stranded helical filaments that assemble into higher-order three-dimensional structures, such as bundles and branched networks [5]. The length of filaments varies from dozens of nanometers (e.g., in branched networks) to several dozen micrometers (e.g., in stress fibers, filopodia, and stereocilia) [12,13,14,15,16]. Together, they form a continuum of systems enabling the reception and transduction of mechanical stimuli across the cell and providing mechanical support for the shape and polarity of cells. The necessity of forming higher-order actin structures is dictated by the immense variety of cellular functions supported by the actin cytoskeleton and a broad range of mechanical forces required to carry them out. Forces generated by these higher-order actin networks, ranging from piconewtons to nanonewtons, aid in cell migration and invasion, internal vesicle movements, endocytosis, exocytosis, phagocytosis, and in cell division [17,18,19].

Actin bundles are linear arrays of actin filaments crosslinked by one or, more often, several different actin-bundling proteins (Figure 1A,B). The length and width of such bundles, and the number of filaments present in them are dictated by their unique set of proteins and by kinetic conditions under which these bundles are formed [20,21], giving each bundled complex a specific structure with different mechanical properties [19,22], Figure 2, Table 1. There are two main types of actin bundles, with either parallel or mixed polarity filament orientations. In parallel (or uniform polarity) actin bundles, the filaments are ordered with consistent polarity, allowing them to conduct work (e.g., membrane deformation) due to the directional F-actin elongation. Parallel actin bundles are present in chemosensory and mechanosensory cell protrusions (microvilli) of most cell types [22,23], stereocilia of inner ear hair cells [24], bristles in the thorax of *Drosophila melanogaster* [25], and in ectoplasmic specializations of Sertoli cells (Figure 2, [26]). Filopodia, microspikes, focal adhesions, and distal ends of dorsal stress fibers, found in most cell types, consist also of parallel actin bundles (Figure 2, [10,27]). These bundles create the force for localized membrane protrusions, while helping cells to resist compressive forces from the membrane [28]. They facilitate cell movement in response to extracellular stimuli or intracellular signaling [29]. The length (1 to 100 μm) and number (one to hundreds) of these bundles per cell, their diameter and the number of actin filaments in them (a few to ~1000) vary depending on the cell type and the structures they support (microvilli, stereocilia, bristles, filopodia, (Figure 2, Table 1), and invadosomes (podosomes/invadopodia) [19,25,27,30,31,32,33,34]. More than two actin-bundling proteins are often associated with these actin structures. Filopodia, thin, actin-rich plasma-membrane protrusions, help in chemosensing during cell migration, wound healing, and cell adhesion to the extracellular matrix [33]. They are enriched in fascin, but additional bundling proteins, such as α-actinin, fimbrin/plastin, filamin, and espin are also present in them under certain conditions [33,35]. Podosomes are actin-based dynamic structures near the plasma membrane of various cells (such as monocytic, endothelial, and smooth muscle cells). They contribute to cell migration, matrix invasiveness, bone remodeling, and mechanosensing [34,36,37] and contain fascin, L-plastin (a hematopoietic cell-specific plastin isoform), and α-actinin [34,38,39,40]. Invadopodia are functionally and structurally similar to podosomes of normal cells, but they are present in tumor cells [34]. Additionally, podosomes and invadopodia are unique in terms of their ability to degrade ECM material by locally releasing proteolytic enzymes [34]. Ectoplasmic specializations of Sertoli cells, that are hybrid testis-specific cell–cell contacts (contributing to the blood–testis barrier), contain espin and T-plastin (a most abundant plastin isoform found in cells of most solid tissues) [19,41,42,43]. Microvilli—the finger-like projections on the surface of several types of cells—increase the total cell surface without substantially increasing its volume [19,23]. Most microvilli contain I-plastin (a plastin isoform found in the intestinal and kidney microvilli and stereocilia of the inner ear), small isoforms of espin, and villin. Similarly, stereocilia are the modified microvilli that transduce mechanical signals into stimulus-dependent electrical signals. They predominantly contain I-plastin and fascin, but also have isoforms of espin and other proteins expressed at smaller quantities [44,45].

In antiparallel (or mixed polarity) bundles, actin filaments are typically organized by bundling proteins with a longer distance between their actin-binding domains (e.g., α-actinin). Therefore, these filaments are typically packed less densely than in parallel bundles, leaving space for the intercalation of myosin thick filaments. Mixed polarity bundles are the constituents of stress fibers of non-muscle cells, myofibrils in muscle cells, cytokinetic contractile rings, and the cell cortex (Figure 2). The filament-forming motor protein myosin II is usually associated with these actin bundles and enables their contractile functions [46,47]. Stress fibers are the primary mediators of cell contraction in non-muscle cells, governing some of their vital processes, including migration, adhesion, and mechanosensing. Although α-actinin is the major actin-bundling protein identified in stress fibers, other bundling proteins (fascin, espin, fimbrin/plastin, and filamin) can also be present (Table 1, [47,48]). A likely role of fascin and espin in such bundles is the stabilization of a subset of filaments with uniform polarities; fimbrin/plastin, despite their compact size, can directly stabilize antiparallel actin assemblies [49,50]. Based on their origin, subcellular location, and protein composition, stress fibers can be grouped into four classes: ventral, dorsal, transverse arcs, and perinuclear actin caps (Figure 2).

Ventral stress fibers are >2 µm long thick actomyosin bundles, whereas dorsal fibers are shorter (~1 µm), devoid of myosin, and unable to contract [51]. These fibers usually contain 10–30 actin filaments per cross-section [47,52]. Ventral stress fibers anchor to focal adhesions from both ends, while dorsal stress fibers anchor to them only from one end, with the other end being embedded into transverse arcs. Ventral stress fibers are the most abundant contractile structures in the cell. They are made of bipolar actin fibers that display alternating patterns of myosin- and α-actinin-enriched regions. In migration, they help retract the motile cells’ trailing edge and establish the cells’ front–rear polarity [53]. Transverse arcs are curved, thin actin filament bundles with repeated α-actinin–myosin patches formed just behind the lamellipodia [47]. They are mainly involved in the persistence of cell motility by acting as a link between the lamella and the focal adhesion-connected dorsal stress fibers [54]. The perinuclear actin cap is a contractile structure (surrounding the nucleus) that emanates from focal adhesions at the leading edge. It influences the nucleus shape and position in the cell by transducing environmental signals to the nucleus via LINC complexes [55,56].

Myofibrils of cardiac and skeletal muscles are assembled into highly organized periodic structures (sarcomeres; Figure 2). In contrast to that, myofibrils in smooth muscles are less ordered and more reminiscent of the ventral stress fibers of non-muscle cells. However, these three types of myofibrils share a striated pattern when immuno-stained with antibodies against myosin II and α-actinin. Each sarcomere contains thick (myosin) filaments in the center and is flanked by regions that contain thin filaments (actin filaments decorated by tropomyosin and troponin) (Figure 2). Actin filaments are embedded with their barbed (plus) ends in the Z-band regions separating sarcomeres and containing α-actinin as the major crosslinking protein (Figure 2, [57]). Therefore, while notably more ordered and with a better-controlled filament length, sarcomeres are organized similarly to stress fibers, which may serve as precursors of sarcomere assembly [58].

In cytokinetic contractile rings of dividing cells, actin filaments form bundles at the division plane (Figure 2) along with the intercalated myosin II filaments. The motor activity of myosin drives the contraction and separation of cells into two daughter cells. α-Actinin, fimbrin/plastin, and anillin are the bundling proteins identified in these structures [59,60,61]. In addition to these actin-rich structures, actin bundles are also present in the cell cortex (Figure 2). As a continuous layer of actin and non-muscle myosin II-enriched networks under the cell membrane, the cell cortex is clearly detectable in rounded mitotic or amoeboid cells. Its thickness varies from ~190 nm in human mitotic cells to 4 µm in some oocyte cells [62,63]. The actin-bundling proteins associated with the cell cortex include α-actinin, fimbrin/plastin, fascin, and filamin [62].

## 3. Actin-Bundling Proteins

Cells contain diverse, complex, and highly regulated actin bundles that form unique molecular architectures associated with specific cellular functions. Accordingly, more than 10 families of actin-bundling proteins have been identified. Physiological actin bundles often acquire their properties from combinations of actin-bundling proteins regulating their steric organization and turnover dynamics. While it is clear that different actin-bundling proteins are required for stabilizing different types of actin bundles, the need for the presence of several types of these proteins in the same bundle is less obvious.

In *Drosophila* bristles [64] and in vertebrate microvilli and stereocilia [65], more than one type of actin-bundling protein is typically involved in forming thick bundles. Despite being partially redundant [66], such bundlers have unique and important functional roles. Mutation or knock-out of any one of them often results in catastrophic cellular effects [64,67,68,69]. Thus, the presence of I-plastin in stereocilium bundles enables their less orderly (random liquid) packing, granting more extensive contact areas between actin fibers and the membrane, and allowing stereocilia to grow to a greater diameter [44].

Although different actin-bundling proteins have distinctive properties, they share some standard features, such as bivalent actin binding. These proteins have either tandem actin-binding domains (as plastin, fascin, and espin), or they form dimers/oligomers (as α-actinin and filamin) with a single actin-binding domain per each monomer to tether actin filaments into bundles [70,71]. Another intriguing property of these proteins is that they typically saturate actin at less than equimolar ratios (typically 1:4–1:13) [19,70,72,73], which is dictated by their divalent binding mode and geometric constraints imposed by the filaments’ helical periodicity. At their maximum actin saturation, the bundling proteins tend to form actin bundles. At a low bundler/actin molar ratio, filament branching is favored, resulting in the formation of meshes and networks, as it has been shown for α-actinin [74,75]. Similarly, filamin forms orthogonal networks at low filamin/actin molar ratios and more compact bundles at their high molar ratios [76]. Thus, increasing the ratio of bundling proteins to actin can convert actin networks into tight bundles, likely contributing to filopodia bundle formation in branched lamellipodial networks.

In addition to F-actin-binding/bundling domains, most bundling proteins also contain other functional domains (such as the LIM, ankyrin, WH2, EF-hand, spectrin repeats, etc.). These domains contribute to the proteins’ structural properties, spacing, flexibility, abilities to interact with G-actin, and their regulation through interactions with other proteins.

### 3.1. α-Actinin

α-Actinin is a medium-sized, dimeric crosslinking protein involved in actin bundling and membrane anchoring of the actin cytoskeleton [77]. α-Actinin has been identified in protists to mammals, but not in plants [78]. An exception from this rule is an Australian tree malletwood (*Rhodamnia argentea*) that encodes a classical α-actinin protein, perhaps acquired via horizontal gene transfer. α-Actinin is a promiscuous actin crosslinker that can crosslink both parallel (e.g., in focal adhesions) and oppositely oriented actin filaments (e.g., in Z-disks of striated muscle, in the contractile ring, and in stress fibers of migrating cells) [72,79]. The mixed-polarity bundles typically contain filaments of myosin II—a primary factor in bundle self-organization that allows force generation for cell spreading, trailing edge retraction in migrating cells, and the contraction of sarcomeres and contractile rings [48,80,81,82]. Moreover, α-actinin acts as an anchorage and scaffolding protein that mediates interactions of cytoskeletal regulatory proteins (e.g., capping protein (CP), vinculin, zyxin, integrin, cell surface receptors, etc.) with the cell membrane [83]. Similarly, as a scaffolding protein, α-actinin coordinates the organization of various actin-based structures, from sarcomeres to focal adhesions [84]. Mutations in α-actinin isoforms are linked to several human diseases, such as autosomal-dominant congenital macrothrombocytopenia [85,86], dilated or hypertrophic cardiomyopathy [87], familial focal segmental glomerulosclerosis (FFSG) [88,89], carcinomas [90,91], and immunological diseases [92].

*Isoforms.* At least six α-actinin isoforms, encoded by four genes (ACTN1-4), have been identified in mammals. ACTN2 and ACTN3 encode muscle-specific Ca^2+^-insensitive isoforms α-actinin-2 and α-actinin-3, while ACTN1 and ACTN4 produce both Ca^2+^-sensitive (ubiquitously expressed) and Ca^2+^-insensitive (expressed in smooth muscle and neuronal cells) isoforms α-actinin-1 and α-actinin-4, which are vital for cytokinesis, cell motility, and spreading [93,94]. α-Actinin-1 is the more ubiquitous of the two non-muscle isoforms, and it is associated in most cells with focal adhesions, perinuclear actin caps, stress fibers, branched cortical cytoskeleton, and contractile rings. In addition to these locations, α-actinin-4 is also found in the nucleus, where it serves as a co-activator for transcription factors, estrogen, and glucocorticoid receptors, MEF2, and RARα [95,96]. Mutations in α-actinin-1 lead to focal segmental glomerulosclerosis [89] and congenital macrothrombocytopenia [97,98]. This correlates with the knock-down of this protein, preventing megakaryocyte polyploidization that is essential for platelet production [86]. Mutations in ACTN4, encoding α-actinin-4 that binds actin more strongly than the wild-type (WT) protein, cause familial focal segmental glomerulosclerosis [88,99].

Among muscle-specific isoforms, α-actinin-2 is abundantly expressed in skeletal and heart muscles but is also found in the brain. α-Actinin-3 is much more specific and is present only in Type 2 fast glycolytic muscle fibers. Its presence is associated with the high performance of elite sprint athletes [100]. Both isoforms are involved in actin filament bundling and anchoring in Z-disks of sarcomeres [101], but also in the scaffolding of many Z-disk components, signaling proteins, and metabolic enzymes. Differences in the scaffolding properties of the two isoforms are believed to account for the switch between aerobic and glycolytic metabolic phenotypes [78].

*Domain organization.* α-Actinin isoforms have high sequence and structural similarity [102]. All isoforms function as homodimers arranged in an antiparallel fashion (Figure 3). Each monomer has a molecular mass of ~110 kDa and spans a distance of ~36 nm (Figure 3, [72,103]), which closely matches the long-pitch repeat of actin filaments. Each monomer consists of the N-terminal actin-binding domain (ABD), connected via a flexible neck region to the rod domain containing four spectrin-like repeats, and a C-terminal regulatory calmodulin-like (CaM) domain with two pairs of EF-hand motifs (EF1-4) (Figure 3, [104,105]). Spectrin repeats in the rod domains of two α-actinin monomers interact extensively to form a stable antiparallel dimer (K_d_ is ~10 pM) [106,107,108,109]. This arrangement places the two ABDs on opposite ends of the α-actinin dimer, at a 90° angle and ~36 nm apart from each other (Figure 3, [72]), which explains the relatively loose arrangement of actin filaments in α-actinin-crosslinked bundles. The CaM domain of one monomer interacts with the “neck region” of the other monomer in the homodimer (Figure 3, [109]). The ABD of α-actinin contains tandem calponin-homology (CH) domains, with two major actin-binding sites (ABS2 and 3) spanning across both CH domains and the third conserved region, identified as a potential ABS (ABS1), being hidden between those domains (Figure 3, [110,111,112]).

*Bundling mechanism.* Similar to other tandem CH domain proteins (e.g., utrophin, spectrin, plastin), the ABD of α-actinin was proposed to exist in equilibrium between its closed and extended conformations [113,114], with its CH1 and CH2 domains either in tight contact with each other or separated, respectively. CH2 is believed to play the role of a negative steric regulator of binding to actin. ABD and F-actin interactions are described using a two-step binding mechanism, in which ABD in a closed conformation binds weakly to F-actin. Following that, actin-induced conformational changes in ABD lead to its transition to extended conformation (i.e., CH2 moves away from CH1). This removes the steric hindrance created by CH2 (in a closed conformation) and thereby enhances its actin binding. However, the extended α-actinin’s ABD conformation was only detected so far in cryo-electron microscopy (cryo-EM) reconstructions of 2D arrays of full-length α-actinin [113] and in the ABD-decorated actin filaments [115]. In the first case, the extended conformation was artificially stabilized due to domain swapping with the neighboring α-actinin molecules. In the second case, the extended conformation (in a 1.6 nm-resolution structure) was deduced from a partial density of CH2 that was inconsistent with the “closed state” of actin-bound ABD. The two-step binding model is indirectly supported by a higher affinity binding to actin of α-actinin-4 mutants with a compromised CH1–CH2 interface [114]. Yet, one such mutant (K255E) was found in the canonical closed state, based on both its X-ray structure and solution-state analysis [114]. Therefore, while there is an agreement that some CH1-CH2 rearrangements are required for binding to actin, the extent and exact nature of these rearrangements remain to be established.

As other bundling proteins, α-actinin binds and bundles F-actin cooperatively [116], consistent with filaments proximity, alignment, and spacing being the main factors in their effective bundling. The apparent K_d_ for actin binding by full-length α-actinins from different organisms varies in the 0.4 to 2.7 µM range [72]. However, the affinity of individual ABD of α-actinin for actin is very low (unmeasurable K_d_ for WT ABD, and K_d_ within the 25–35 µM range for the high-affinity mutants of α-actinin-4 [114]). This suggests that the simultaneous binding of both ABDs of the dimer contributes to this interaction, resulting in higher affinity of full-length α-actinins for actin. Two-dimensional arrays of full-length α-actinin assembled on a lipid monolayer showed extreme variability of the bridges, with angles at which α-actinin-2 crosslinks actin filaments in the bundles clustered at 60°, 90°, and 120°, both in parallel and antiparallel arrangements [103]. In addition to that, binding of dimers to the same actin filament [0° and 180° angles) was also observed [103]. The necessity for binding of both α-actinin domains to actin (either within the same filament or between filaments in a bundle) for gaining a measurable affinity explains the low (1:14) stoichiometry of α-actinin binding to actin [72], which correlates with the number of actin subunits (~13) in the long-pitch turn of the actin helix.

The extreme variability of α-actinin’s bridge angles (enabled by a flexible linker of 25–30 residues between ABD and the rod domain) defines the remarkable polymorphism of α-actinin crosslinked actin assemblies. Overall, α-actinin–actin bundles have distorted square lattices (i.e., diamond-shaped lattices having inter-axial angles of ~80°/100° instead of 90°/90°) (Figure 1C), with the filaments roughly 35 nm apart in a meshwork-like topology (branched topology) [117]. The porosity and compressibility of such meshworks depend on their ratio of actin to α-actinin [75], but also on the kinetic parameters under which the bundles/meshworks are assembled [20].

*Regulation.* Calcium binding inhibits the bundling activity of Ca^2+^-sensitive splice variants of non-muscle α-actinin (-1 and -4) by inducing conformational changes in the CaM domain, which is in proximity to ABD of a partnering subunit in the dimer [109,118]. Upon Ca^2+-^ binding, EF1-2 motif induces conformational changes in EF3-4 that reinforce its binding to the neck region. This limits the flexibility around the neck region, causing steric clashes between ABDs of partnering subunits of a dimer, and inhibits bundling [109,118]. In contrast to that, EF1-2 motifs of the muscle isoforms (α-actinin-2 and -3) are not regulated by Ca^2+^. Instead, in muscle-specific α-actinins, phosphatidylinositol bis- and tris-phosphate (PIP2 and PIP3) binding to ABD releases EF3-4 from the neck, enabling its interaction with titin [109,119,120,121], a giant protein integrating the thin and thick filaments of striated muscle sarcomeres. Titin binding further favors the flexibility of the neck region, allowing ABD domains to acquire the orientation necessary for crosslinking of overlapping parts of actin filaments in the Z-disk (in antiparallel orientation) [109]. Moreover, phosphorylation of specific residues on human α-actinin-1 (Y12) and α-actinin-4 (Y4, T32, and Y265) [122,123], and limited proteolysis of chicken muscle actinin by calpain-1 and calpain-2 [124,125] also contribute to the regulation of its binding/bundling activity. Thus, α-actinin isoforms are regulated at several different levels, resulting in the functional flexibility needed for fulfilling their numerous cell- and tissue-specific functions.

*Protein partners.* In addition to crosslinking F-actin, α-actinin interacts with several other cytoskeleton and membrane-associated proteins. As mentioned above, the muscle isoforms α-actinin-2 and -3 work as scaffolds for integrating the activity of many Z-disk components, signaling proteins, and metabolic enzymes. α-Actinin interactions with titin are important for the proper functioning and assembly of sarcomeres [126]. These interactions are highly regulated by PIP2 binding to EF3-4 motifs of α-actinin, which acts as the docking site for titin’s Z-repeats. These Z-repeats are ~45 a.a. long regions, and their number correlates with the number of α-actinin dimers bound to actin bundles in the Z-disks of various types of striated muscle [109,127,128,129]. At focal adhesions, α-actinin associates with zyxin, vinculin, and with cell surface receptors β-integrins [86,130]. PDZ- and LIM-domain protein CLP36 binding to α-actinin-1 shifts its localization from focal adhesions to stress fibers [131]. Conversely, α-actinin promotes cell migration by activating the MEKK1/calpain pathway, which inhibits focal adhesion formation by cleaving vinculin and talin [132]. α-Actinin also promotes cell migration through its interactions with dynamin-2 and HAMLET [133,134]. Thus, α-actinin is a connecting link between transmembrane receptors and the cytoskeleton. It helps in transducing external signals to the cytoplasm and in regulating actin reorganization per cell requirements. Independent of its cytoskeleton remodeling roles, α-actinin-4 regulates gene expressions upon its translocation to the nucleus, where it interacts with several nuclear receptors, chromatin remodeling proteins, and transcription factors [90,135]. Altogether, α-actinins are a multifaceted family of proteins that link external stimuli and gene regulation to cell migration and proliferation.

### 3.2. Fimbrin/Plastin

Fimbrins (plastins in animals) are highly conserved actin-bundling proteins (~70 kDa) identified in all eukaryotes, including yeast [136,137]. They are localized mainly at the cell’s edge, in focal adhesions, ruffling membranes, lamellipodia, filopodia, microvilli, and stereocilia [138,139]. In yeast, they are present predominantly in endocytic actin patches and cytokinetic rings [49].

*Isoforms.* Fimbrin was first identified and purified from chicken enterocytes [136]. Antibodies against intestinal fimbrin recognized a related protein at the edge of other non-muscle cells, giving origin to the protein name [from Latin “*fimbria*”, i.e., edge]. In parallel, a hematopoietic isoform was initially identified in cancer, but not in normal cells, and named “plastin” as related to neoplastic transformations [140,141]. Notably, the name “fimbrin” is used less commonly to define vertebrate members of this protein family. This is because plastins had been recognized earlier to include three different isoforms in all vertebrates. Three homologous isoforms of plastins (I, L, and T for **i**ntestinal, **l**eucocyte- and solid **t**issue-specific) are expressed in mammals in a tissue-specific manner, with distinct roles in actin filament organization [137,142]. Plastins I, L, and T are also known as plastins 1, 2, and 3 (PLS1-3), respectively. Accordingly, in the HGNC database (HUGO Gene Nomenclature Committee at the European Bioinformatics Institute), genes encoding I- and T-plastins are listed as *PLS1* and *PLS3*, while the L-plastin (or plastin 2) gene is listed as *LCP1* (for lymphocyte cytosolic protein 1), as it was first described in 1982 as a lymphocyte-specific product [143]. I-plastin (PLS1) is found in the intestinal and kidney microvilli and stereocilia of the inner ear. L-plastin (PLS2, LCP1) is exclusively present in hematopoietic cell lineages under normal conditions, but is expressed ectopically in many malignant tumors [144]. T-plastin (PLS3), the most abundant isoform, is found in cells of most other tissues [145]. Fimbrin/plastin participates in many processes, including endocytosis, cell motility, cell adhesion, mechanotransduction, Ca^2+^ homeostasis, vesicle trafficking, and axonal local mRNA translation [11].

In humans, mutations in I-plastin are linked to autosomal dominant hereditary deafness [146]. I-plastin knock-out mice have shorter and thinner stereocilia and develop moderate hearing loss [147]. While intestinal microvillus bundles can form without plastin, it is required for terminal web assembly via its interaction with keratin [148] and, together with two other actin bundlers, villin and espin, it is essential for apical retention of proteins involved in intestinal physiology [66].

L-plastin supports the migration and invasive ability of various blood cells and contributes to the stability of the T-cell immune synapse and macrophage podosomes. It promotes platelet formation in a miR-125a-5p-dependent manner and enhances NLRP3 inflammasome assembly [149,150,151,152,153]. This isoform is linked to non-Hodgkin lymphoma [154], whereas its ectopic expression is associated with a high invasiveness in solid tumors [144,155].

T-plastin is expressed in most solid tissues, and, accordingly, it is the least specialized of the three isoforms. During embryogenesis, it is expressed also in the intestinal and inner ear hair cells, where it is later replaced by I-plastin [156]. Located in the X-chromosome, the *PLS3* gene is associated with several X-linked congenital disorders. Its deletions and several missense or short insertion mutations (without frame shift) are linked to congenital osteoporosis, likely due to the anomalous (reduced or elevated) sensitivity to Ca^2+^ of the respective mutant proteins [68]. A separate set of mutations results in a severe congenital diaphragmatic hernia (CDH) [157]. While the effects of CDH mutations have not been biochemically characterized, they may be related to the role of T-plastin as a regulator of basement membrane assembly and epidermal morphogenesis [158]. Interestingly, T-plastin is recognized as a protective modifier ameliorating symptoms of other congenital diseases. High levels of T-plastin suppress the symptoms of spinal muscular atrophy (SMA) in a subset of patients by acting upon endocytic pathways in the affected neurons [159,160]. In a zebrafish SMA model, T- and L-, but not I-plastin, decreased the phenotype severity [161], emphasizing isoform specialization, but also their partial redundancy. While L-plastin is associated with solid tumors, ectopic expression of T-plastin is a marker and indicator of poor prognosis of blood malignancies [162,163] and pancreatic cancers [164,165]. The role of T-plastin in various cell functions has been recently reviewed [11].

*Domain organization and function.* Unlike all other tandem CH domain proteins, fimbrin/plastin is a monomeric protein comprising the N-terminal Ca^2+^-binding regulatory domain with two EF-hand motifs, followed by two closely arranged actin-binding domains (ABD1 and ABD2) assembled in a core domain (Figure 3, [166]). This unique tight arrangement of ABDs helps in the formation of densely packed fimbrin/plastin–actin bundles (Figure 1B,C) and enables allosteric regulation of the domains’ activity [167]. Full-length fimbrin/plastin has never been crystallized, most likely due to a long flexible linker connecting its EF-hands with the core domain. The structure of the core domain is known for yeast and plant fimbrins [168], but not for animal plastins. Each ABD contains two tandem CH domains that interact with subdomains 1 and 2 of adjacent actin subunits in the filament [167,169,170].

Despite having homologous CH domains, ABD1 (CH1-2) and ABD2 (CH3-4) show very different affinities for actin [167,170,171,172]. ABD2 of human L-plastin binds F-actin very tightly, with a low-nanomolar K_d_, and potently nucleates actin filaments [167]. In cryo-EM images, it shows ordered and stoichiometric binding to actin [170,172]. The nucleation ability of ABD2 is blocked effectively by the ABD1 fragment added in trans, as the affinity of the two ABDs for each other is also in the low nanomolar range, similar to that of ABD2 for actin. In striking contrast to that, the binding of ABD1 to actin is weak (micromolar K_d_) and similar to that of full-length plastins [171]. Its actin decoration—as seen by cryo-EM—is partial and polymorphic, unless stabilized by covalent crosslinking to actin [167]. Both ABDs interact with actin in a similar mode, but the footprint of ABD2 on actin is bigger, reflecting its higher affinity (Figure 3]. The location of EF-hands relative to the ABD core is uncertain, but they are predicted to be docked at the loop-rich region between CH2 (ABD1) and CH3 (ABD2) (Figure 3, [170]). In the presence of Ca^2+^, the EF-hands of plastin share a substantial similarity with CaM and bind to a switch helix (homologous to canonical CaM-binding motifs), which is located in the linker segment connecting the regulatory and core domains (CBM in Figure 3, [171,173]).

*Bundling mechanism.* Despite a tight arrangement of its ABDs, fimbrin/plastin can crosslink actin filaments in both parallel and antiparallel arrays [49,50], which correlates with its dual localization in microvilli and stereocilia (parallel bundles), and in contractile rings and the cell cortex (mixed bundles). In 2D arrays, plastin crosslinks actin filaments into dense parallel bundles with a ~120 Å [12 nm) inter-filament distance, and it is spaced after every ~13.5 actin monomers (Figure 3, [169]). Such versatility is achieved by the separation of ABD1 and ABD2 upon binding to actin, which is possible due to the flexibility of its interdomain linkers [50,167]. Based on a recent machine-learning-enabled cryo-EM structure analysis, it has been proposed that ABD2 binds to the first actin filament and forms a metastable single-filament-engaged pre-bundling state [50,172]. This binding partially releases interdomain inhibition and re-orients ABD1 for a second filament binding, either in a parallel or antiparallel orientation [50,172]. However, a discrepancy as to which ABD domain binds first persists. Biochemical studies suggested that ABD1 is always accessible for actin filament binding (and it binds first), and, instead, ABD2 binding to actin is regulated by the autoinhibition due to ABD1 and Ca^2+^ binding [167,170,171]. Interestingly, it was proposed that the ABD1 of T-plastin may be sufficient to affect actin turnover, stabilization, and assembly independently of the bundling activity [139], most likely by competing with ADF/cofilin. However, the human L478P T-plastin variant, which does not bundle actin due to a mutation in its ABD2, leads to osteoporosis and localizes diffusely in the cytosol [68]. These observations suggest that bundling is the primary and essential function of this protein.

*Regulation.* Actin bundling (but not binding) activity of human plastins (in all isoforms) is inhibited in the presence of Ca^2+^, but the range of Ca^2+^ sensitivities differs among them [171,174,175,176]. It is proposed that, in the presence of Ca^2+^, ABD2 is masked by the Ca^2+^-binding regulatory domain and only ABD1 is available for actin filament binding [170]. According to this hypothesis, this constraint is released when Ca^2+^ dissociates from the regulatory domain, and ABD2 becomes available for filament binding [170]. Most intriguingly, the inhibitory allosteric interaction of ABD1 with ABD2 enables fine-tuning of the plastin’s crosslinking strength as the release of this inhibition by a phospho-mimicking S406E mutation (reproducing a physiologically relevant modification) converts human L-plastin into a highly potent bundling protein, essentially resistant to Ca^2+^. It is speculated that the tunable allosteric design of the actin-binding core gives rise to the functional versatility of this protein—via condition-dependent stabilization of morphologically distinct actin bundles and meshes with different sensitivities to Ca^2+^. This may explain the otherwise unfavorable localization of the protein in association with poorly aligned meshworks of the cell’s edge [167]. Fimbrin/plastin Ca^2+^-dependent actin-bundling functions have been recently reviewed by Schwebach et al. [68] and Wolff et al. [11]. The regulation by Ca^2+^ is not universal, as fimbrins from *Schizosaccharomyces pombe*, *Arabidopsis thaliana*, and *Tetrahymena* are reported to bundle actin filaments in a Ca^2+^-insensitive manner [177,178,179].

*Protein partners.* In fission yeast, fimbrin is associated mainly with actin patches involved in endocytosis. Tropomyosin (Cdc8) is another actin-binding protein that stabilizes F-actin and helps in maintaining actin cables and contractile ring assembly during cytokinesis. Moreover, it protects F-actin against cofilin-mediated severing [180,181]. Fimbrin has been shown to competitively inhibit the binding of tropomyosins to F-actin and help in cofilin-mediated severing. This, in turn, improved bundling in the in vitro total internal reflection fluorescence (TIRF) microscopy experiments due to the increased mobility of severed filaments and their incorporation into nearby bundles [181,182]. This also allows actin patch recycling and maintenance during endocytosis in yeast cells [49,182]. This mechanism balances the distribution of actin between pools of branched (Arp2/3 complex controlled) endocytic networks and linear (formin- and tropomyosin-controlled) actin cables [49,181,182]. Altogether, fimbrin/plastin functions extend beyond actin bundling, as it also regulates actin dynamics that remodel actin structures according to cell requirements. In addition to actin filaments, fimbrin/plastin interacts with intermediate filament proteins, such as vimentin and keratin, which are components of the cell-adhesion-related intermediate filaments [148,183].

### 3.3. Fascin

The fascin family of ~55 kDa evolutionary conserved globular proteins forms ordered and rigid parallel actin bundles (Figure 1B, [184,185]). These bundles provide cell stability, elasticity, and the pushing force needed during cell adhesion, migration, sensing, and invasion. Therefore, fascin–actin bundles are highly localized in filopodia, lamellipodia, microspikes, invadopodia, and stress fibers of motile cells [186,187,188]. The fascin–actin bundles’ involvement in polar body extrusion and spindle migration during meiosis [189], as well as their regulation of nuclear actin dynamics, the nucleolus, and chromatin modifications, has been well established [190,191,192].

*Isoforms.* Fascin was first identified in sea urchins [193] and later found to be universally present in most other eukaryotic organisms [190,194,195,196]. Humans have three fascin isoforms encoded by three different genes. Fascin-1 is expressed at high levels in neurons, endothelial and mesenchymal cells, and at very low levels in normal epithelial cells [197]. Fascin-2 is present in retinal photoreceptors and inner ear stereocilia [198,199,200]. Fascin-3 is present specifically in mature spermatozoa, where it may contribute to microfilament rearrangements that accompany fertilization [201,202]. Yet, it is dispensable for spermatogenesis and fertility in mice [203]. Fascin-3 is the most diverse isoform that shares only 28% identity and 43% similarity with the other two isoforms (which share 72% similarity). Despite this moderate similarity, the residues involved in stabilizing the core and putative actin-binding sites are conserved [190,204]. In *Drosophila*, a fascin homolog “*singed*” is vital for bristle formation and border cell migration during egg fertilization [194,205].

*Structure and bundling mechanism.* Fascin has four tandem β-trefoil folds (β-T, Figure 3) arranged in a pseudo-two-fold symmetry, yielding three putative actin-binding sites (ABS) (Figure 3, [73,185,204]). Fascin’s ABS1 (located between the β-T1 and 4) and ABS2 (formed by β-T1 and 2) are present on the same side of the protein. Fascin’s ABS3 is on the opposite side, in a β-T3, and interacts with another actin filament (Figure 3, [185,206]). Fascin cooperatively binds (K_d_ ~150 nM; [196,207]) and bundles (K_d_ ~270 nM; [116]) actin filaments and crosslinks them with high efficiency in a calcium-independent manner [204]. The incorporation of fascin during elongation of bundles constrains their flexibility and leads to a discrete bundle geometry, as opposed to when it is added to preformed long filaments, which instead merge into interconnected networks [21]. When fascin is added to short parallel filaments, it crosslinks them tightly and favors unidirectional bundle elongation. During the formation of filopodia, this unidirectional actin polymerization-mediated push is needed to overcome the compressive force of the cell membrane. In bundles, actin filaments are shifted relative to each other by 2.7 nm (an axial rise per actin subunit) and are bound at an average angle of 61°, resulting in an ordered hexagonal structure ([73,185,206], Figure 1C). The consequent inter-filament distance of ~11 nm, due to the small molecular size of fascin, gives rise to compact and dense bundles [73,185,204,206]. Cryo-electron tomography (cryo-ET) of filopodia bundles (100–200 nm thick) showed >30 actin filaments typically present in them [206].

*Regulation.* Fascin’s function is regulated by several factors, such as post-translational modifications and protein–protein interactions. Phosphorylation of human fascin-1 at Y23, S38, S39, and S274 regulates its actin-binding/bundling activity and its interactions with other proteins [7,207,208]. The above residues are highly conserved across fascin isoforms, among which S39 is the site of phosphorylation-mediated inhibition of all fascins. Furthermore, human fascin-1’s acetylation at K471, or mono-ubiquitination at K247 and K250, adversely affects its actin-bundling activity [8,209]. Actin filament decoration with either tropomyosin or drebrin inhibits fascin binding and bundling activities [210,211].

*Protein partners.* Fascin regulates the activity and binding of other actin-binding proteins. It inhibits myosin II activity by inhibiting its binding to bundled actin in vitro and in vivo [212,213], and alters cofilin’s binding kinetics [214]. Fascin directly or indirectly cooperates with other proteins and protein assemblies, including microtubules [208], formins (Daam1 and FLMN3), Ena/VASP [215,216,217], and Rab35 [218]. Moreover, fascin also regulates cell motility in an actin-independent manner, via direct interactions with microtubules [208]. Fascin overexpression is identified in several cancers and is connected to increased host invasiveness, metastasis, and mortality [219,220,221]. Thus, in recent years, fascin has become a key prognostic marker and drug target for cancers [222].

### 3.4. Espin

Espin is a unique family of bundling proteins with no substantial sequence identity in its actin-bundling domain with other actin-binding proteins. The long isoform, later designated espin-1, was first identified in parallel actin bundles of ectoplasmic specializations of Sertoli cell–spermatid junctions in rat testis [26]. Espin’s small isoforms were found later in microvilli-like protrusions of kidney and intestinal epithelia, as well as in those of various chemo- and mechano-sensory cells, such as stereocilia of cochlear and vestibular hair cells [223,224], taste cell receptors [225], vomeronasal sensory neurons [225], tactile epithelial Merkel cells in the skin [225], microvilli of spiral ganglion neurons [226], and in dendritic spines of cerebellar Purkinje cells [227]. Espins are enriched in stereocilia actin bundles, and their mutations are associated with deafness and vestibular dysfunction in mice and humans, respectively [228,229,230]. Genetic deletion of espins in mice leads to thinning and shortening of hair cell stereocilia, as well as to their region-dependent degeneration and collapse [229]. In stereocilia, espins are distributed uniformly along the bundle, near the cell membrane, suggesting their role in stabilizing the membrane–cytoskeleton connection [231]. In striking contrast to that, the long isoform of espin is localized exclusively in the tips of stereocilia—the only region where active actin dynamics are observed.

*Isoforms.* Four espin isoforms (espin-1 to espin-4) of notably different molecular mass (from 110 kDa to 25 kDa, in order of decreasing size) have been identified in mammals, all originating from a single gene. All isoforms share the C-terminal actin-bundling domain but vary in their N-terminal regions that contain other functional domains. Espin-1, the longest isoform (~110 kDa), was first characterized in junctional plaques of Sertoli cell ectoplasmic specializations [26], giving it the name espin (for **e**ctoplasmic **sp**ecialization + **-in**). Later on, a short espin isoform (<30 kDa) was identified in microvilli of the intestine and proximal renal tubule [232]. Espin-1 is unique in containing ankyrin repeats (AR) at the N-terminus (Figure 3). Intermediate-size isoforms, espin-2 and espin-3, have two splice variants referred to as espin-2A and -2B and espin-3A and -3B, respectively [224].

All espins are expressed in a tissue-specific and developmentally regulated fashion, often with more than one espin isoform co-existing at any given time [224,226,233]. Thus, while both espin-2 and espin-3 isoforms are detected in developing rat cochlear as early as embryonic day 10 (E10), their fate is different. Espin-2 levels steadily decrease towards birth, and it almost disappears at postnatal day 15 (P15). In contrast to that, espin-3 levels rise throughout gestation and in early prenatal stages. The appearance of espin-1, the largest isoform, coincides with birth (P0), and it becomes the dominant isoform at P15, when espin-4 (the shortest isoform) is also detected at high levels [233]. Similarly, a mature retinal sensory epithelium contains espin-1, -3, and -4, while the vomeronasal organ sensory cells (vestigial in humans but present in many other mammals and in reptiles) are enriched in espin-2 and -3, and contain trace amounts of espin-1, i.e., representing the developmental isoform distribution of the cochlear epithelium [224,233,234]. The differential expression of all isoforms suggests their unique role in the formation and length maintenance of stereocilia and microvilli-like protrusions [225,234]. *Drosophila*’s *forked* proteins show 35–39% sequence identity with espins. They are involved in forming small, disordered actin bundles beneath the plasma membrane, which act as initiation sites for forming actin bundles for bristle development [19,235].

*Structure and bundling mechanism.* Espins are monomeric proteins with C-terminal actin-binding domain (ABD) responsible for a potent actin-bundling activity, suggesting the presence of at least two actin-binding sites within ABD [232]. In cells, ABD alone is sufficient to cause notable elongation of microvilli in immortalized kidney epithelial cells [234]. Furthermore, high espin levels correlate with longer stereocilia in cochlear cells [232]. Yet, due to the absence of espin’s 3D structure, the mechanisms behind actin binding and bundling by this domain are not known. All isoforms contribute to the elongation, but shorter isoforms (espin-3 and -4) tend to produce long and thin protrusions, which contrast with shorter and thicker microvilli stabilized by the longer (espin-1 and -2) isoforms [234]. While shorter isoforms decorate the entire stereocilium length, as detected by pan-espin antibodies, espin-1 is localized at the tip of these protrusions in a myosin IIIA-dependent manner, contributing to the proper organization of at least a subset of the inner ear staircase stereocilia assemblies [236,237].

All espins contain a WASP homology 2 domain (WH2] (Figure 3). This domain is known to bind actin monomers and/or filament barbed ends [238]. Large espins (espin-1 and -2), have additional actin-binding sites and two proline-rich regions that can bind profilin (Figure 3, [26,223,239]). In vivo, large espins are reported to increase in bundle diameter by assembling additional layers of actin filaments at their periphery [234,240], which can be due to the additional actin-binding sites present in them. Espin-3 has one proline-rich domain region, while espin-4 has none (Figure 3). The proline-rich regions of actin-binding proteins are known to associate with profilin-bound G-actin and are often utilized by the actin machinery to raise the local concentration of polymerization-competent actin to fuel actin filaments elongation [241]. Interestingly, deletion of either WH2 or the proline-rich domain does not affect espin’s elongation activity but eliminates the highly dynamic (diffusible) actin component in stereocilium FRAP experiments [234]. Given that F-actin dynamics in stereocilia are limited to their tips, the presence of this sequestered G-actin population in the body of the bundle is puzzling. It may account for the need for bundle repair to address the sound wave-related mechanical damage. Furthermore, as proline-rich regions of many proteins are known to participate inprotein–protein interactions in many signaling cascades [242], this region may play a regulatory role in espin activity.

Espin-1 differs from other espin isoforms by containing eight N-terminal ankyrin-like repeats (AR) (Figure 3, [26]) that mediate its translocation to the stereocilium tip via interactions with the myosin IIIA motor [236,243]. This interaction, as well as the interaction of ankyrin repeats of espin-like protein (ESPNL, a negative stereocilia length regulator lacking the actin-bundling domain), is critical for the regulation of stereocilia staircase length and fine spacing [236]. Moreover, the AR region shields additional actin-binding sites of espin-1, keeping it in its autoinhibited state (less active state) [239]. This autoinhibition is also relieved by binding myosin III, further substantiating the regulatory role of myosin III (via espin-1) in stereocilia formation [239].

As espins show sequence variations mainly at their N-terminus, these regions are likely the site of interaction with other proteins that are implicated in cytoskeleton remodeling and external signal transduction. In addition to espin-1 interaction with myosin IIIA [237,243,244], the N-terminal proline-rich peptide of espins in Purkinje cells interacts with IRSp53 (insulin receptor substrate protein 53 kDa), which is an adapter protein that links membrane-bound small GTPases with cytoplasmic effector proteins, and is known to regulate the actin cytoskeleton [227,245]. The N-terminus variation of espin isoforms gives them a unique identity and functional role, explaining the need for their spatiotemporal expression during development [233]. Although espins amount to only ~15% of the total bundling proteins in stereocilia (likely due to their localization at the periphery of the bundle), mutations in espin result in stereocilia degeneration [44,45,229]. Overall, espins are multifaceted proteins involved in different processes of bundle formation in microvilli-like protrusions, including their initiation, crosslinking, lengthening, and thickening.

Espins bind actin with high affinity (*K*_d_ = 10–100 nM) and bundle filaments at lower molar ratios (~1 espin per 20–50 actin monomers) than most of the other bundling proteins [26,223,232]. They bind at the lateral surface of F-actin, with a stoichiometry of ~1 espin per 4–6 actin monomers [26,223]. The resulting espin–actin bundles are uniformly oriented, well-ordered, and dense. In these bundles, actin filaments are hexagonally coordinated, with a ~12 nm inter-filament distance ([44,246,247], Figure 1C).

*Regulation.* Espin bundles actin in a Ca^2+^-independent manner, in contrast to other bundling proteins present in stereocilia and microvilli (α-actinin, villin, and fimbrin/plastin) [223,232]. This characteristic of espin is crucial for forming stable actin bundles and maintaining stereocilia and microvilli structures in sensory cells, as they are open to local influx of Ca^2+^ during signaling [248]. Espin isoforms espin-1, -2, and -4, but not -3, bind PIP2 [224] due to the presence of the respective PIP2-binding domains in the longer isoforms, and the unique N-terminal domains in espin-4 produced due to alternative splicing (Figure 3). These domains may regulate espins association with the membrane and their actin-binding/bundling activity, similarly to other actin-bundling proteins [109]. As PIP2 is a signaling lipid, it is possible that espin can regulate actin bundle formation directly in response to extracellular signals. Espin’s activity is also regulated by its transport to the tips of stereocilia by myosin IIIA [237,243], which is the key element in giving stereocilia a tapered appearance [229,236,237]. A mutually exclusive interaction of espin with actin or whirlin (see below) is believed to contribute to the regulation of espins.

*Protein partners.* Besides its interaction with actin, myosin, and profilin, espin interacts with IRSp53, which recruits actin elongators, enabled/vasodilator-stimulated phosphoprotein (Ena/VASP), and diaphanous formin mDia1, thus assisting in bundle elongation [227,245]. When not bound to actin, espin also interacts with whirlin, a protein whose mutations cause retinal degeneration and hearing loss [249]. Adequate levels of both proteins are essential for the length and thickness homeostasis of stereocilia [249]. Although a direct interaction between espin and Cobl (Cordon-bleu WH2 repeat protein) has not been demonstrated, as far as we know, a co-expression of these two proteins in mouse melanoma cells results in the formation of numerous lamellipodia-like protrusions that are absent in the cells expressing either protein alone [250].

## 4. Assembly and Disassembly of Actin Bundles

Actin bundles are the main components of several complex cell structures (both stable and transient), such as microvilli, stereocilia, filopodia, and stress fibers. While all these structures are made up of packed actin bundles, they differ considerably in their molecular composition, dynamic nature, and biogenesis [225]. To form actin bundles, their assembly has to be initiated at precise locations and proceed in coordination with ever-changing internal and external factors in a tractable manner [251]. Once the assembly begins, each of the involved proteins’ levels, kinetics of their interactions with actin and with each other, and their regulation states contribute to the unique bundle’s structural features (e.g., number, length, and rate of filament elongation) and functions. Given the variety of unique structures and compositions of different bundles, the precise mechanisms of bundle assembly vary substantially, and their detailed analysis is beyond the scope of this review. Instead, we focus on some general features and common mechanisms shared by most of them and the key functional properties of the contributing proteins. Here we integrate the available information to shed light on the complex mechanism of bundles assembly in microvilli, stereocilia, and filopodia. Figure 4 illustrates the role of different actin-binding and anchoring proteins controlling the formation of these structures. For information on bundle assembly in stress fibers and invadopodia, the readers can turn to related publications [12,34,47,252,253,254].

### 4.1. Initiation

The first step in filopodia/microvilli/stereocilia bundle assembly involves its “foundation”. Tilney and colleagues described the bundle founding region as microvillus—a small, dense patch on the plasma membrane from which actin filaments start to emerge [22]. These patches consist primarily of actin and actin regulatory proteins that regulate the barbed end’s dynamics, such as Ena/VASP and formins (actin nucleation and elongation promoting proteins), capping proteins, profilin, and myosins [30,35]. Since formins and Ena/VASP are bound to the membrane, actin filaments in such bundles elongate by adding monomers at the membrane-associated (barbed) ends of actin filaments [255]. The signal for membrane-associated bundle formation is typically triggered by a Rho-GTPase-mediated signaling cascade acting on the nucleation and elongation-promoting factors [256,257,258].

Although filopodia are one of the most ubiquitous and best characterized actin-containing systems, our understanding of their assembly initiation is limited. Two distinct models of the mechanism of filopodial bundle nucleation have been proposed [35,52]. The convergent model suggests that parallel actin bundles originate from pre-existing Arp2/3 branched actin networks in lamellipodia [13]. In this model, formins and Ena/VASP are the proteins that reorganize the Arp2/3 complex-induced actin networks into parallel actin bundles [259,260]. The alternative, a nucleation model, suggests that actin bundles are formed by a de novo mechanism mediated by formins attached to the plasma membrane. It is possible that both of these models function in the cell in parallel, and depending on cellular conditions one is preferred over the other. The convergent model seems to describe better bundles in which mixed polarity of filaments is needed, while the nucleation model is best suited for parallel bundles formed in filopodia or microvilli [261,262].

### 4.2. Elongation

Once the foundation is set, the filaments in bundles grow from their barbed ends facing the membrane [255]. The force generated by elongation of actin bundles drives the membrane deformation, resulting in protrusions. Formins [263] and Ena/VASP proteins [264], the well-recognized elongation factors that promote processive elongation of F-actin at the barbed ends [264,265], are the main drivers of this step. However, whether Ena/VASP proteins participate in this process as bona fide elongation factors are still debatable. Some studies suggest that the primary function of Ena/VASP proteins is to replace the barbed end capping protein (CP), which otherwise blocks the elongation [260]. In turn, Ena/VASP can be displaced by formins to drive the barbed end’s elongation [266,267]. In neurons, mDia2 [268] and Daam [217,269] formins are shown to play a pivotal role in filopodia assembly.

Actin elongation in bundles can be continuous (single filaments in bundles growing continuously) or discontinuous (fusion of short filaments/bundles to form long bundles). Thus, during the initial stage of bristle formation in *Drosophila*, the nascent actin bundles form discrete discontinuous modules separated by gaps. In the later stages, a continued elongation links these short bundles to each other in a head-to-tail fashion, resulting in overlapping “graft” regions at the junction due to the bundles extending one over the other. These “grafts” are subsequently filled by the elongation of actin filaments, thereby merging the adjacent short bundles into smooth and continuous actin bundles [270]. In contrast to that, actin bundles in microvilli appear to assemble from filaments that grow continuously, with their barbed ends attached to the membrane, yielding long, continuous actin filaments [22]. The mechanisms of bundle nucleation and elongation, the length of the bundles, and the number of filaments in them are not obviously dictated by the type of bundling proteins involved [22,234].

Intriguingly, in stereocilia, neither formins nor Ena/VASP are identified at a significant level. It appears that after initiation of stereocilia formation from a patch referred to as microvillus, the Ena/VASP function may be taken over by myosins (myosin IIIA, IIIB, VIIA, and XVA), espin-1, and other proteins (whirlin, harmonin, and Eps8). These proteins are shown to play a key role in bundle length and the maintenance of staircase patterns in stereocilia [236,243,271,272,273,274,275,276,277]. However, a recent study suggests that myosin XVA—in its nucleotide-free state—can nucleate actin filaments at the tip of stereocilia, by mediating F-actin inter-subunit contacts [278]. Further studies are needed to understand the mechanism by which myosins regulate stereocilia formation. In *Drosophila* bristles, myosin XVA is shown to play a crucial role in bundle length and shape regulation [279], while other contributing proteins and processes are yet to be identified.

Bundle length is also regulated by extracellular signals (from growth factors, pH, cations, mechanical stress, etc. [280,281,282,283]). Actin bundles are directly or indirectly associated with transmembrane receptors (Eps8, TRPC, etc. [282,284]) or transmembrane proteins (cadherin-23 and protocadherin-15 [283]) at the tips of cell structures, such as microvilli, stereocilia, and filopodia. These proteins help in transducing mechanochemical signals and regulate actin bundle growth [281,282,284].

### 4.3. Crosslinking

Actin polymerization, up to a filament length of ~0.7 µm, generates ~1 pN (piconewton) of force, after which actin filaments start to buckle due to the compressive force from the membrane [285]. The generated force is insufficient for a sustainable “membrane pushing”, as needed during cell motility or for maintaining stable actin structures (microvilli and stereocilia) [285]. The role of actin-bundling proteins in motility was clarified in the studies of *Listeria monocytogenes* comet tail. Thus, the removal of Arp2/3 complexes (which initially provided branched actin networks for generating the force for propelling the bacterium) did not impede the motility as long as fascin–actin bundles were also present [286]. Similarly, linear bundles of actin filaments can generate enough force to extend the protrusion required for cell migration and impart stiffness and stability to cell structures [287].

All known physiological bundles contain several actin-bundling proteins (with specific properties) that cooperate in providing distinct molecular architectures and functions. In the *Drosophila* bristles, fascin and forked (a homolog of espin protein) control actin bundle organization to maintain the correct morphology of bristles [25,64]. Forked protein starts the initial bundling, which is later taken over by fascin [64,288]. Fascin converts the loosely packed nascent assemblies into compact and rigid bundles [289]. A similar mechanism may work in filopodia formation in mammalian cells, in which plastin, localized at the cell’s periphery (via unknown mechanisms) initially tethers actin filaments into bundles that are then stiffened by fascin [188]. Similarly, in stereocilia, fascin-2 is expressed at the later stage of actin bundles formation, while different isoforms of espin and plastin are initially present [200,290,291,292]. This observation supports the role of espin and plastin in bundles initiation and elongation, while fascin imparts stiffness and stability to bundles structure.

Furthermore, actin-bundling proteins spatiotemporally regulate the structure of bundles by cooperating with some and competing with other actin-bundling proteins (loose bundles with α-actinin and dense bundles with fascin, fimbrin/plastin, and espin) [116,293]. Hence, α-actinin is restricted in cells to the basal portion of filopodia, while fascin and plastin are present along its entire length [294]. Similarly, in fission yeast, fimbrin competes with α-actinin for binding to actin filaments and supports the formation of actin patches instead of contractile rings [295]. The mechanisms behind the segregated binding of bundling proteins involve both altering the topology and arrangement of actin filaments in the bundles. For instance, fascin binding over-twists the filaments in bundles, limiting the number of α-actinin binding sites [116,296]. Conversely, cooperative binding of two or more different bundling proteins is also possible. Thus, the bundles supported only by fascin are theoretically limited in size to ~20 actin filaments due to the imposed structural constraints [296]. In cells, actin–fascin bundles with >20 filaments are present, suggesting that other actin-bundling proteins help to overcome these structural limitations. Overall, the segregated/cooperative binding of bundling proteins helps to maintain specific actin structures in cells. The molar ratio of individual bundling proteins to actin affects the bundles’ crosslinking density, shape, and mechanical properties [69,188].

The properties of bundling proteins determine the number of filaments in the bundles and, thereby, their width [297]. For example, espin directly regulates the width and length of actin bundles in stereocilia, giving them an asymmetric, tapered appearance [228,229]. Not only the bundling protein’s nature but also the timing of its addition affects the bundle width and polarity [292,298,299]. Fascin’s addition to long filaments generates thin and flexible bundles, in contrast to its addition during filaments nucleation [21,300]. Moreover, the presence of other actin-binding or bundling proteins also modulates bundle thickness [188,214].

Bundling proteins also facilitate bundles elongation by “collaborating” with actin filaments elongating proteins (Ena/VASP and formins) [215,301,302]. The “alliance” between Ena/VASP and formins with fascin appears crucial for uniform elongation of all filaments in actin bundles in filopodia [215,301,302]. Some formins (that exist as dimers [303]) and Ena/VASP proteins (that exist as tetramers [302]) contribute to bundles formation as they have multiple actin-binding sites, a property common in all actin-bundling proteins. In addition, these proteins can translocate from the barbed ends to the side of actin filaments, which aids their crosslinking activity [304,305,306].

Overexpression of bundling proteins (e.g., plastin and fascin) is often associated with retarded bundles depolymerization [179,290,307,308], suggesting that the crosslinking of filaments stabilizes their assemblies and shifts the balance towards actin polymerization. This has a positive effect on the overall rate of bundles elongation and maintenance. The presence of α-actinin is also shown to limit the myosin-mediated contraction strength of stress fibers via filament stabilization [309]. However, even these stabilization effects are not straightforward, as in microvilli the bundling protein espin is associated with a mild acceleration of treadmilling [234]. Certain bundling proteins facilitate myosin binding to actin and help to transport cargo at the tip of actin protrusions and to regulate their length [244,274,310]. On the other hand, the binding surface of plastin’s ABDs on actin overlaps with that of myosin motors, suggesting a tentative competition between these proteins. Similarly, Arp2/3 complexes that are extensively associated with branched filament networks inhibit fascin binding to actin filaments (this promotes parallel bundles formation [311]), thus ensuring the presence of one type of actin complex structure in the cell at a given site at any given time. Overall, in addition to directly shaping the bundles by crosslinking actin filaments, bundling proteins contribute broadly to the regulation of each step of bundles formation.

### 4.4. Disassembly/Severing

The formation, shape, size, and composition of numerous actin bundles, often simultaneously present in the cell, are highly tuned to specific cellular functions, suggesting the existence of regulatory mechanisms that intricately control actin-rich protrusions. For instance, the defined length and thickness of microvilli and stereocilia are critical for their absorptive and mechanosensory functions and are determined by a fine balance of assembly/disassembly that is unique for each protrusion type.

Actin treadmilling is the key element of actin filaments’ dynamic behavior. During treadmilling in vitro, ADP-G-actin is released from the pointed ends of aged filaments at a fixed rate of 0.25 s^−1^ [312,313], converted to ATP-G-actin, and reused at the barbed ends. In a test tube, treadmilling maintains the overall filament length; in cells, global treadmilling enables constant reorganization of individual actin filaments and larger assemblies in response to cellular needs. In vivo, the treadmilling rate is not limited by the pointed ends’ depolymerization rate, as it can be dramatically accelerated by actin-binding proteins. In LLC-PK1-CL4 kidney epithelial cells, parallel bundles in microvilli treadmill at a rate of ~3 s^−1^, which is ~2.5 times slower than the average treadmilling rate in stationary filopodia [234,314]. Remarkably, treadmilling is characteristic for microvilli and filopodia but is almost absent in stereocilia [314,315,316,317,318]. Despite earlier reports on active treadmilling in inner ear stereocilia, most recent studies agree on the lack of (or extremely slow) treadmilling in stereocilia bundles, where actin dynamics are limited instead to the filament barbed ends at the distal tips [316,318,319]. These unusual actin dynamics is likely dictated by the need for a precise regulation of the staircase stereocilia length at different levels, which is achieved during development and at the early postnatal stages. Therefore, treadmilling rates differ substantially in a cell-type- and bundle-type-specific manner.

Under non-regulated conditions, the rates of actin polymerization depend on actin concentration, while depolymerization rates do not. These differences are more pronounced in the presence of polymerization-assisting actin-binding proteins (e.g., formins and profilin), which increase the total and local G-actin concentration and thus accelerate filament elongation [320,321,322]. A discrepancy between slow filaments depolymerization and fast elongation is compensated in cells by ADF/cofilin and its partner proteins. These proteins help in increasing: [1] the number of depolymerizing ends by severing filaments, and [2] the rates of subunit dissociation via direct influences on the filaments’ ends. These proteins also promote ADP- to ATP-G-actin conversion to replenish the polymerizable actin pool. To slow down treadmilling, actin dynamics at the filaments’ ends can be regulated by barbed end (CP) and pointed end (tropomodulin) capping proteins. Below, we briefly discuss actin-binding proteins involved in filament disassembly, along with their role in this process.

#### 4.4.1. ADF/Cofilin Protein Family

ADF and cofilins (~17–19 kDa) belong to the unique family of proteins with the ability to accelerate severing/depolymerization by changing filament geometry (e.g., twist and inter-subunit contacts) [323]. Another key property of these proteins is their higher affinity for aged, ADP-enriched actin filaments, enabling discriminative destabilization of older filaments that completed their cellular roles. As compared to some other proteins (particularly from the gelsolin family), cofilin is recognized as a relatively weak severing protein. ADF/cofilins promote severing and depolymerization via their cooperative binding to actin filaments that alters their twist and rearranges interprotomer contacts [324,325,326]. Interestingly, the severing ability of ADF/cofilins is optimal at sub-saturating concentrations, as higher protein levels stabilize rather than sever the filaments [324,327]. Microfluidics-assisted TIRF experiments showed that at saturating concentrations cofilin accelerates the disassembly of actin filaments from their barbed ends more than from the pointed ends, resulting in similar depolymerization rates for both ends [327,328]. Furthermore, cofilin-promoted barbed end depolymerization is “unstoppable”, as filament capping by CP is inhibited by cofilin [329], paradoxically resulting in faster barbed end depolymerization in its presence. Therefore, depending on its concentration, ADF/cofilin can either potentiate filament/bundle treadmilling (i.e., depolymerization at the pointed end and re-polymerization at the barbed end) or cause their complete depolymerization. The former scenario is relevant to dynamically stable protrusions (e.g., microvilli or persisting filopodia), while the latter may be essential for the removal of transient structures (e.g., transient filopodia). The actual role of ADF/cofilins in filament/bundle turnover is even more complex due to its cooperation with CAP1/2 (cyclase-associated proteins 1/2], which can dramatically accelerate depolymerization from the filaments’ pointed ends [330]. Fast bundle/filament turnover is associated with rapid cell motility [331,332]. Cofilin’s role in cell motility—the generation of an actin monomers pool—has been extensively reviewed [331,333,334].

Cofilin collaborates with fascin and fimbrin/plastin in bundle severing/disassembly [181,214,335]. For fascin, the joint action of the two proteins can be partially explained by the hyper-twisting of actin filaments upon cofilin binding to already strained fascin–actin bundles [214,296]. By pre-straining the filaments and limiting their ability to compensate for cofilin-induced twisting, fascin reduces the amount of cofilin required for their effective severing. The severing, if not followed by complete depolymerization, may result in the generation of new filaments barbed ends being used as a template for new filaments formation, and thus increase bundles thickness [214]. In cells, cofilin appeared at the tips of retracting, but not protruding, filopodia and partially overlapped with fascin upon their shortening, suggesting that the two proteins may indeed cooperate in the disassembly of physiological bundles [214]. Fimbrin/plastin promotes filament severing by cofilin using a different mechanism. While fimbrin’s and cofilin’s binding sites on actin overlap, which should result in their mutual inhibition, fimbrin also competes effectively with tropomyosin. Tropomyosin is an abundant coiled-coil protein, and one of its main functions in the cell is to stabilize actin filaments by protecting them from the destabilizing effects of ADF/cofilin [181]. Notably, microvilli bundles contain fimbrin/plastin but not tropomyosin [136,336].

The activity of ADF/cofilin in animal cells is universally regulated by LIM (double zinc finger **L**in11, **I**sl- 1, and **M**ec-3] and TES (testicular protein) kinases-mediated phosphorylation of its Ser3, which reduces its affinity for actin filaments [337,338,339]. Together with cofilin phosphatases, chronophin and slingshot [339], LIM and TES kinases determine the active dephosphorylated cofilin concentration in the cell. The cytoplasmic concentrations of ADF/cofilin can be reduced also by their sequestering at the membrane via a cooperative interaction with PIP2 signaling lipids [340]. Other actin-binding proteins, including actin-interacting protein 1 (Aip1) [341,342], coronin [342], and cyclase-associated protein (CAP1/2) [330,343], also regulate cofilin’s severing and/or depolymerization activity.

#### 4.4.2. Capping Proteins

Capping protein (CP or α/β CapZ heterodimer) binds to actin filament barbed ends and inhibits their elongation and shortening. Actin filaments in bundles of uniform length (such as those present in stable structures of microvilli, stereocilia, and Z-disks of sarcomeres) are regulated by CP [30,292,344]. CP is present at high concentrations at the leading edge of moving cells, where it blocks the polymerization of a subset of actin filaments by competing with formins and Ena/VASP at the filaments’ barbed ends. By limiting the number of free barbed ends (i.e., avoiding futile elongation cycles), CP helps to replenish the actin pool in dynamic actin structures, allowing effective and productive elongation of the uncapped ends [345]. Accordingly, CP depletion in cells impedes cell motility [346]. In cells, the effective concentration of CP is controlled by CP-sequestering protein myotrophin/V-1 [347] and by CARMIL (capping protein regulator and myosin 1 linker) [348], allowing it to regulate the balance between branched and linear actin assemblies [349].

Recent studies showed the regulation of CP binding to filaments barbed ends by ADF/cofilin [328] and a related protein twinfilin [345,350,351]. Twinfilin reduces the depolymerization rate of ADP-actin, while accelerating the depolymerization of ADP-P_i_-actin, resulting in similar rates of depolymerization of old and new filaments from their barbed ends [345,350]. This mechanism has the advantage of limiting productive actin polymerization to a select number of filaments in the Arp2/3-branched networks, while also providing fresh G-actin for their elongation. Twinfilin may play a similar role at the stereocilia tips as its overexpression (or knock-out of CP) correlates with reduced stereocilia length and degeneration [344,352]. Moreover, twinfilin is also localized at the tips of filopodia [353], lamellipodial networks [350], and yeast cortical actin patches [354], which are the sites of high actin dynamics [355].

Gelsolin is one of the most abundant and potent capping and severing proteins that act in a Ca^2+^-dependent manner [356]. Its binding disrupts interprotomer actin interactions resulting in filament severing. After severing, gelsolin remains attached to the barbed ends of filaments and blocks their elongation [357]. Interestingly, gelsolin is related to villin, one of the three major actin bundlers in the intestinal brush border microvilli. These two proteins share the Ca^2+^-dependent severing/capping activities, but gelsolin does not have the bundling ability of villin. Knock-down of gelsolin in mice leads to defects in cellular motility, stereocilia development, and in platelet function during blood clotting [358,359].

#### 4.4.3. Profilin

Profilin is a ubiquitous actin-monomer-binding protein with a high affinity for ATP-actin (K_d_ = 0.1 µM). It stimulates actin elongation by two mechanisms: (1) by catalyzing the exchange of nucleotides on G-actin and (2) by delivering ATP-actin to the poly-proline sequences of actin elongation factors at the barbed ends of actin filaments. Importantly, it effectively blocks the nucleation and pointed end elongation of filaments, supporting unidirectional actin assembly. This is mainly due to the ability of profilin to block the pointed end elongation of actin filaments, which was thought to be impossible under physiological conditions in non-muscle cells. Most recently, however, processive polymerization at the actin filaments’ pointed ends was discovered to be facilitated by *Vibrio* protein toxins VopF and VopL, both in living cells and in vitro [360]. At high concentrations, combined with low concentrations of monomeric actin, profilin slows the elongation of filaments by promoting dissociation of the terminal actin monomers [241,361]. Thus, in coordination with CP and a G-actin-sequestering small protein thymosin-β4, profilin helps to maintain the ATP-actin pool (up to ~100 μM) [46,362,363]. The binding of profilin to poly-proline regions of large espins contributes to maintaining high levels of G-actin in stereocilia bundles, whose roles are yet to be fully understood.

It is to be noted that the role of profilin in nucleotide exchange in G-actin has been called into question. It is now suggested that CAP (cyclase-associated protein) actually performs this function [364]. Further studies are needed to verify the role of these proteins in G-actin nucleotide exchange in vivo. To summarize, profilin plays a vital role in actin bundles by maintaining the local pool of actin monomers and supporting unidirectional filament elongation at the tip of the bundles.

#### 4.4.4. MICALs

MICALs (microtubule-associated monooxygenase, **ca**lponin, and **L**IM domain containing proteins) are a family of actin regulatory redox (oxidation-reduction) enzymes conserved from *Drosophila* to humans [365,366]. They are essential for cellular navigation, axonal guidance, motility, and bristle development [367,368,369]. NADPH and FAD are the co-factors for their oxidation activity [370,371]. MICAL binds to actin filaments and oxidizes two methionines (Met 44 and Met47) in the D-loop of actin [372]. This post-translational modification disrupts the inter-subunit interactions in the affected filaments, leading to their structural destabilization and the disassembly of MICAL-oxidized actin (Mox-actin) filaments [370,371,373]. MICAL also binds directly to bundled actin filaments, irrespective of the bundling protein present on them, and disassembles them [371]. Mox-actin has a high critical concentration for actin polymerization [374] and a poor ability to form bundles (our unpublished data). MICAL oxidation of actin filaments also potentiates cofilin binding and thereby facilitates their rapid disassembly [374].

#### 4.4.5. Myosins

Myosins are a large superfamily of actin-dependent, ATP-driven, force-producing motor proteins required for cell movement [375]. A filamentous myosin II stabilizes bundles in contractile structures of antiparallel stress fibers [376]. In contrast to that, in the filopodia of neurons, myosin has been shown to promote bundle severing, mainly by producing kinks and buckles in parallel bundles through its contractile activity [377]. Recently, gelsolin has been shown to stimulate myosin severing activity, providing a more efficient severing than by either of these proteins alone [378]. A key function of non-filamentous myosins is the transport of cargo to specific cell locations using actin filaments as tracks [375]. Overall, different types of myosins play different roles during actin bundle formation. Thus, a fine balance between their activity and localization is needed for the maintenance of proper bundle length and shape. The role of myosins in filopodia, microvilli, and stereocilia has been extensively reviewed [277,310].

In addition to the above discussed proteins, numerous other actin partners are involved in the assembly and regulation of actin bundles. Among the most notable, no less than 40 human tropomyosin isoforms orchestrate the dynamics and accessibility of actin to severing and treadmilling [379]. In coordination with tropomyosin, tropomodulin caps and halts the dynamics at filaments pointed ends [380]. Talin, vinculin, and many other adaptor proteins that link actin bundles to cell membranes in focal adhesions, invadosomes, tight junctions, and the cell cortex are discussed elsewhere [12,34,47,62,252,253,254].

## 5. Future Directions

Despite a large amount of information available on this topic, a quantitative understanding of actin bundles formation is yet to be achieved. While proteins involved in the development of physiological bundles have been mainly identified due to advances in mass spectrometry and related techniques, the time and sequence of their appearance in the bundles and their precise functions are yet to be clarified. Thus, despite the abundance of proteomic data on actin-rich unique structures, such as stereocilia, microvilli, and filopodia [30,45,381], the mechanisms that determine their precise length and width are unknown. Recent advances in various modifications of proximity labeling (mainly, proximity biotinylation) by engineered promiscuous enzymes [382] opened new and unprecedented possibilities for spatial and temporal identification of partner molecules. Understanding of actin assemblies is yet to benefit fully from these powerful techniques and their forthcoming modifications.

Similarly, the structures of the majority of actin bundles are not understood in depth. The last decade’s progress in the resolution of cryo-EM and cryo-TM methods has helped in determining the structure of bundled actin and provided information on the constituents present at the bundles’ tips [50,206]. These techniques have produced previously unprecedented analysis of giant actin-based assemblies of podosomes [383]; other actin-based super-structures will undoubtedly follow. Super-resolution fluorescence microscopy—in all its variants [384]—and ultrastructure expansion microscopy [385] are other fascinating tools that have added to—and will advance—our knowledge of cell assemblies. In addition to classical NMR and protein X-ray crystallography approaches, their modern modifications, such as magic angle spinning (solid-state) NMR (MAS NMR), have been successfully applied to characterize protein-decorated actin filaments [386,387], enabling the evaluation of protein dynamics. Adaptation of such structural methods to applications in cells [388] can aid in mapping the structures of proteins in their physiological context. Furthermore, the development of computational algorithms for the effective analysis of data produced by the above methods, most likely with the application of machine learning techniques, should enable progress in these areas. Yet, none of these modern techniques can replace protein characterization using classical biochemistry methods. Contrary to a common misconception, a thorough understanding of the biochemical properties of most proteins, and in particular of their cooperation with each other, is far from being complete. Therefore, a thoughtful and creative combination of modern biochemical, structural, live- and fixed/frozen-cell imaging, proteomics, and other approaches is required for a comprehensive characterization of actin-based assemblies. We can expect the coming decade to be revolutionary in revealing cell secrets.

## Figures and Tables

**Figure 1 biomolecules-13-00450-f001:**
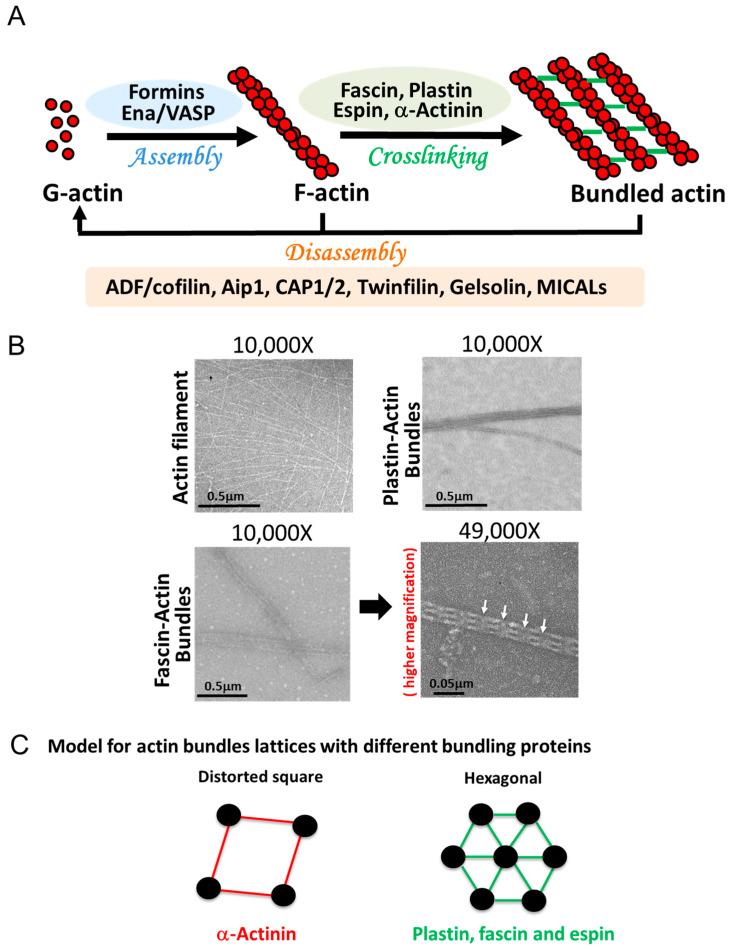
Overview of actin organization in the cell. (**A**) Assembly of actin monomers into linear filaments and higher-order actin structures. Actin monomers bind nucleation and elongation factors, such as Ena/VASP and formins, that assist in these processes, usually near the cytosolic side of plasma membranes. To form actin bundles in a spatially and temporally controlled manner, bundling proteins are recruited to crosslink these filaments. Actin filaments and bundles disassemble into actin oligomers (via severing) and monomers (via accelerated depolymerization) with the help of several disassembly/severing proteins, which contribute to actin turnover in cells. (**B**) EM micrographs of negatively stained actin filaments alone and in the presence of fascin or T-plastin. The magnification of the images is shown on the top of each micrograph. A high-magnification image (0.05 μm) shows an ordered fascin–actin bundle with periodic striations (indicated by arrows). These striations are formed by fascin bound to actin filaments. (**C**) Model of bundle lattices formed in the presence of different bundling proteins discussed in this review. The black circles denote the actin filaments, and the colored lines denote the actin-bundling proteins. Notably, in the case of a hexagonal lattice, the inter-filament distance varies with the size of the bundling proteins involved.

**Figure 2 biomolecules-13-00450-f002:**
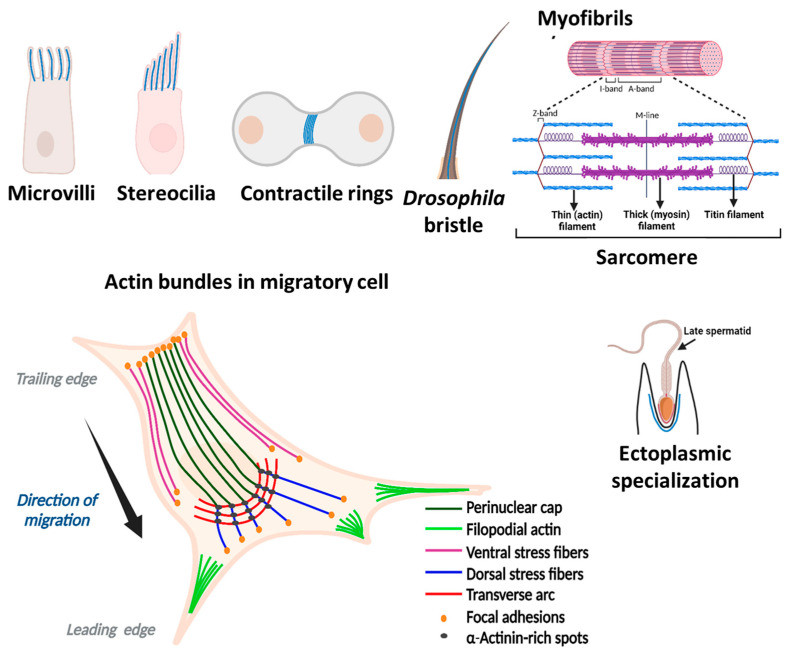
Schematic representation of a cell with different architectures of actin bundles. The blue lines show the actin bundles’ location in different types of cells. In the sarcomere, “thick filaments” (purple) are composed of myosin, while “thin filaments” (blue) are actin bundles decorated with troponin and tropomyosin (decoration is not shown). The different actin bundles structures in migratory cells are shown using different colors. The thick brown line surrounding the perimeter of the cell denotes the cell cortex. This figure was created using BioRender.com.

**Figure 3 biomolecules-13-00450-f003:**
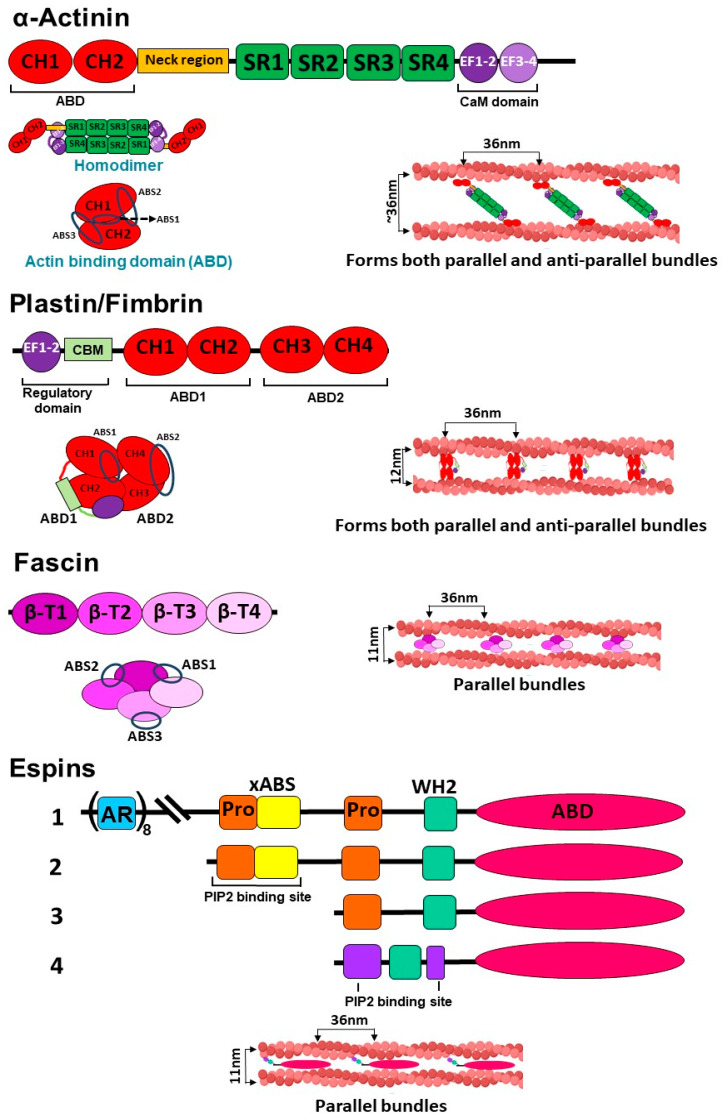
A schematic view of domain organization of actin-bundling proteins and their interactions with F-actin. α-Actinin comprises of a pair of calponin-homology (CH) domains and a calcium regulatory domain with two EF-hand motifs. α-Actinin has one ABD and a rod domain, consisting of four spectrin repeats (SR) engaged in antiparallel dimer formation (homodimer). The flexibility around the neck region determines its Ca^2+^-dependent bundling activity. Plastin/fimbrin contain two pairs of calponin-homology (CH) domains that form two actin-binding domains (ABD), and one calcium regulatory domain with two EF-hand motifs. In plastin/fimbrin the long flexible linker between EF-hand motifs and the CH1 is referred to as CaM/EF-hand binding motif (CBM). Fascin has four β-trefoil folds (β-T) arranged to form three actin-binding sites, two on one side and one on the other side. Four isoforms of espin vary in molecular masses between ~110 and 25 kDa (espin-1 to espin-4). ABD stands for actin-binding domain, WH2 for WASP homology 2 domain, Pro for proline-rich domain, xABS for the additional actin-binding site in large espins. Espin-1 has a unique N-terminus eight ankyrin repeats (AR) sequence. For espin–actin bundles, a possible bundle arrangement is shown in the absence of its structure.

**Figure 4 biomolecules-13-00450-f004:**
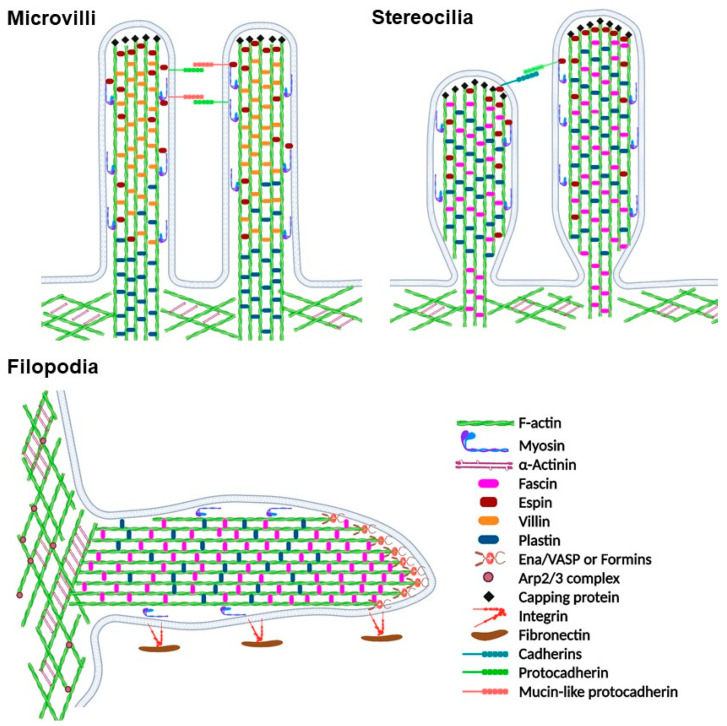
Organization and protein composition of actin-bundle-rich structures. Microvilli, stereocilia, and filopodia share overlapping compositions of proteins with few differences. For instance, microvilli and stereocilia consist of bundling proteins espin and plastin, but the third bundling protein differs between them: fascin in stereocilia and villin in microvilli. Filopodia comprise mainly of fascin and plastin, with α-actinin being present only in their sub-membrane area. In microvilli and stereocilia, the barbed ends of actin filaments are protected by capping proteins, which stabilize the structures. In filopodia, the barbed ends of actin filaments are bound to Ena/VASP and/or formins, which aid in filaments elongation and synergize with bundling proteins, such as fascin or fimbrin/plastin. These variations in protein composition may be responsible for the unique features of these structures (length, diameter, and actin turnover). Moreover, the membrane receptors vary significantly among these structures: protocadherins and mucin-like protocadherins are present in microvilli, while cadherins and protocadherins are in stereocilia. Other proteins (harmonin, Eps8, and whirlin) are also involved but not shown in the figure. In filopodia, integrins act as the transmembrane receptors that interact with fibronectin in the extracellular matrix and transduce the signal for actin remodeling with the help of several other accessory proteins (not shown in the figure). Several classes of myosins (motor proteins) are also associated with bundled F-actin to transport cargo, both anterograde and retrograde, and to perform other functions. In this figure, all myosins are represented similarly. This figure was created using BioRender.com.

**Table 1 biomolecules-13-00450-t001:** Properties of actin bundles in different structures.

Location	Microvilli	Stereocilia	Filopodia	Stress fibers	Bristles (*drosophila*)
**Cell type**	Most cells (frequent in epithelial cells)	Auditory and vestibular sensory cells	Motile cells	Most cells (prominent in fibroblasts, smooth muscle, and endothelial cells)	Sensory organ precursor cells
**Function**	Increase apical surface area for absorption	Mechano-electrical signaling	Sensory and guiding	Contraction and adhesions	Mechanosensing
**Length**	100 nm to 2 µm	1.5–15 µm	≤10 µm	≥2 µm	Macrochaetes: 250–300 μm, Microchaetes: 70 μm (non-continuous 1–5 µm units)
**Diameter**	50–100 nm	~200 nm	20–200 nm	Varies from cell to cell	Varies
**Number of actin filaments**	30–40	~400–3000	10–30	10–30	7–18 bundles with hundreds of filaments
**Actin filament organization**	Parallel (unipolar)	Parallel (unipolar)	Parallel (unipolar)	Mixed (bipolar)	Parallel
**Bundling proteins**	Espin, plastin, villin	Fascin, espin, plastin	Fascin, α-actinin, plastin, espin	α-Actinin, fascin filamin	*singed* (fascin), *forked* (espin)

## Data Availability

Not applicable.

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
