# Peer review of "Actin Bundles Dynamics and Architecture"

_biomolecules, 2023, doi:10.3390/biom13030450_

Round 1

Reviewer 1 Report

This review on actin-filament bundling proteins compiles a large amount of information from 363 publications cited in the References. This impressive work suffers however, in my view, from inconsistencies and unclear passages. I regret that I cannot recommend the manuscript for publication without a substantial revision.

A couple of additional Figures may help understanding the text on specific structures of bundles formed by the various proteins discussed, in particular when angles are given.

There are quite a number of spelling and other minor errors in the table, figures and text.

 I like to specify some of the points where I have problems to follow.

Lines 40/41:          This  statement is too general to be meaningful; what is a structure that is relevant to its specific functions?

Line 57:                 Filaments of dozens of nanometers length: is this correct?

Line 63:                 What is “a broad range of forces”; please specify.

Line 76:                 Do really “most cell types” contain microvilli?

Line 90:                 What are “hybrid cell-cell contacts”?

Line 108:               Gelsolin should be introduced not only as a stabilizer but also as a severing protein.

Table 1:                 Stress fibers /All cells: Do really all small amoeboid cells contain stress fibers?

According to Table 1, plastins are only present in structures with parallel bundles, not in stress fibers. However, line 124 says that plastins can directly stabilize anti-parallel actin assemblies. Is that consistent?

Line 156:               What about anillin in cytokinetic rings?

Lines 295/296:      Please explain: low stoichiometry … correlates with the number of actin subunits.

Line 300:               How does that fit: tight actin bundles and variation of angles?

Line 303:               What are “distorted square lattices” and “a meshwork-like topology”?

Line 334:               What are α-actinin bridges that correlate with titin’s Z-repeats?

Line 373:               What does “axonal local translation” mean?

Line 386:               Change to “T-plastin expressed in most solid tissues is the least….”

Line 423:               Why does not ABD2 determine the affinity of full-length plastin to actin? What does here “polymorphic” mean?

Line 432:               “both parallel and anti-parallel”: not consistent with Table 1 where plastins are only associated with parallel bundles.

Line 416 and 425: “bind in a non-identical manner” and “in a similar mode”. How does that fit together?

Line 428:               Please explain the relevance of calmodulin binding motifs in the present context.

Line 451:               Why does the plastin not bundle when it binds?

Line 463:               What does “otherwise favorable” mean in this context?

Line 467:               “Other fimbrins”: does this mean all fimbrins except human ones?

Lines 485/501:      “Highly conserved” – fascin-3 only 28% identity – does this fit?

Line 509:               In Fig. 2 the three actin-binding sites are not recognizable.

Lines 513/514:      Why a higher affinity for binding than for bundling?

Lines 522/523:      I don’t understand the meaning of two sites and the angles between “each filament”.

Line 524:               “Compact and dense” bundles: what does that mean and how is that compatible with varying angles?

Line 542:               Please explain “can regulate cell motility in an actin-independent manner”.

Line 571:               Ankyrin repeats: are these designated AR in Fig. 2 ?

Line 584:               “and contains”: the retinal epithelium or the vomeronasal organ?

Line 591:               “two conserved actin-binding sites”: In Fig. 2 only one is indicated.

Line 649:               Similar to fascin-actin bundles … are uniformly oriented. In Table 1 and otherwise in the text, fascin results in mixed orientation.

Line 715:               The Tilney reference is missing.

Line 753:               I do not understand how the elongation step smooths the (head-tail?) links.

Line 775:               What does here “Additionally” mean?

Line 815:               “does not support”. In other contexts, the cooperation of different bundles is emphasized.

Line 870:               “fixed rate” – however line 875  3 sec-1 which is slower …

Line 893:               “filament ends”: both ends?

Line 905:               Please specify “weak”: low affinity or slow?

Line 906:               Please specify “cooperative”.

Line 909:               “higher protein levels stabilize”, but line 932 “At high concentration, cofilin, promotes the disassembly…”

Line 970:               “ADP-Pi ; actin”: why not ATP-actin in new filaments?

Lines 1045/1046:  I don’t understand the meaning.

Author Response

We thank the reviewer for the valuable comments. These comments have helped us enhance our manuscript’s quality and readability. The issues pointed out to us are addressed and answered below. Kindly use tracked .pdf version of the manuscript for referring to page and line numbers.

A couple of additional Figures may help understanding the text on specific structures of bundles formed by the various proteins discussed, in particular when angles are given.

We have now addressed the above point by adding a new EM image in Figure 1 and also modified Figure 2 (now Figure 3) with additional information (scheme of domain organization and bundle topology with the four bundling proteins mentioned in the text).

There are quite a number of spelling and other minor errors in the table, figures and text.

As mentioned above, we have thoroughly reviewed the manuscript and corrected all formatting, spelling, and grammatical issues.  

 I like to specify some of the points where I have problems to follow.

Lines 40/41:          This  statement is too general to be meaningful; what is a structure that is relevant to its specific functions?

We thank the Reviewer for bringing this to our attention. We have omitted the entire sentence to avoid confusion (page 1, lines 40-42).

Line 57:                 Filaments of dozens of nanometers length: is this correct?

To the best of our knowledge, this statement is correct, and a respective reference is added to support our statement. The reference added is ref 16.

Line 63:                 What is “a broad range of forces”; please specify.

The range was specified in the following sentence. To clarify, we have now changed the abbreviated units of forces (pN and nN) to fully spelled words (piconewtons and nanonewtons), (page 3, lines 71 - 73).

Line 76:                 Do really “most cell types” contain microvilli?

Yes, microvilli are present in most cell types, albeit in lower densities than in specialized epithelial cells (new ref 23 is added to support our statement).

Line 90:                 What are “hybrid cell-cell contacts”?

Thank you for raising this point. By “hybrid” we mean more than one type of cell-cell contact, such as focal contacts, tight junctions, gap junctions, and desmosome-like junctions present in Ectoplasmic specializations of Sertoli cells. We would like to keep the text as it is; however, we are adding two refs (42 and 43) for further reading and in support of our statement.  

Line 108:               Gelsolin should be introduced not only as a stabilizer but also as a severing protein.

A statement referring to that is now added to the legend of Figure 1A (page 4, line 129).

Table 1:                 Stress fibers /All cells: Do really all small amoeboid cells contain stress fibers?

According to Table 1, plastins are only present in structures with parallel bundles, not in stress fibers. However, line 124 says that plastins can directly stabilize anti-parallel actin assemblies. Is that consistent?

We thank the Reviewer for bringing this to our attention, and now we have specified the type of cells containing stress fibers in Table 1. However, we are correct in stating that stress fibers are devoid of plastin and also that plastins stabilize anti-parallel bundles. Plastins are present in the contractile ring and cell cortex which consist of anti-parallel bundles, as mentioned on page 6, lines 191, 195).

Line 156:               What about anillin in cytokinetic rings?

We have added anillin to the bundling protein list in cytokinetic rings (page 6, line 191).

Lines 295/296:      Please explain: low stoichiometry … correlates with the number of actin subunits.

The sentence was updated in the following way: “The necessity for binding of both α-actinin domains to actin (either within the same filament or between filaments in a bundle) for gaining a measurable affinity explains the low [1:14] stoichiometry of α-actinin binding to actin [72], which closely correlates with the number of actin subunits (~13) in the long-pitch turn of the actin helix. ” (page 11, lines 345-349).

Line 300:               How does that fit: tight actin bundles and variation of angles?

The word “tight” has been removed from the sentence (page 11, line 351).

Line 303:               What are “distorted square lattices” and “a meshwork-like topology”?

These terms are commonly used for describing actin structures and topology. We have provided an explanation for using these specific terms. The sentence is now updated in the following way “Overall, α-actinin-actin bundles have distorted square lattices (i.e., diamond-shaped lattices having inter-axial angles of ~ 80°/100° instead of 90°/90°), with the filaments roughly 35 nm apart in a meshwork-like topology (branched topology) [117].” (page 11, lines 354-356).

Line 334:               What are α-actinin bridges that correlate with titin’s Z-repeats?

We have explained the context of α-actinin bridges and modified the text in the following way “ … and their number correlates with the number of α-actinin dimers bound to actin bundles in the Z-discs of various types of striated muscle [109,127-129].” (page 12, lines 385-387).

Line 373:               What does “axonal local translation” mean?

 “ local translation” is used to describe the “on demand” local expression of mRNA at a specific site; in this case in axons Therefore, we have modified the phrase to “ axonal local mRNA translation” (page 13, line 428). 

 Line 386:               Change to “T-plastin expressed in most solid tissues is the least….”

We have rephrased this sentence in the following way: “T-plastin is expressed in most solid tissues, and it is the least specialized of the three isoforms” (page 13 and line 444).

Line 423:               Why does not ABD2 determine the affinity of full-length plastin to actin? What does here “polymorphic” mean?

This is because ABD2 is potently inhibited by ABD1, as explained in the text (page 14, lines 479-480). “Polymorphic” means bound in several nonidentical conformations, a common term used to describe structures.

Line 432:               “both parallel and anti-parallel”: not consistent with Table 1 where plastins are only associated with parallel bundles.

The sentence “ fimbrin/plastin can crosslink actin filaments in both parallel and anti-parallel arrays" is correct because plastins are present in the contractile rings and cell cortex which consist of anti-parallel bundles (Page 6, lines 191, 195).   

Line 416 and 425: “bind in a non-identical manner” and “in a similar mode”. How does that fit together?

Similar, but non-identical, just as described: “Both ABDs interact with actin in a similar mode, but the footprint of ABD2 on actin is bigger, reflecting its higher affinity”(page 14, line 484-485). Nevertheless, to avoid confusion, we removed the statement about “non-identical ” binding (page 13, lines 476).

Line 428:               Please explain the relevance of calmodulin binding motifs in the present context.

The sentence has been updated in the following way: “In the presence of Ca2+, EF-hands of plastin share substantial similarity with CaM and bind to a switch helix (homologous to canonical CaM-binding motifs), which is located in the linker segment connecting the regulatory and core domains” (page 14, lines 488-491).

Line 451:               Why does the plastin not bundle when it binds?

Because it binds via only one of the ABDs, as it is specified in the next sentence: “It is proposed that in the presence of Ca2+ ABD2 is masked by the Ca2+ binding regulatory domain, and only ABD1 is available for actin filaments binding.” (page 14, lines 505-507). Alternative hypotheses are also provided.

Line 463:               What does “otherwise favorable” mean in this context?

Thank you for noticing this mistake/typo. It was supposed to be “otherwise unfavorable”(page 14, lines 528): “This may explain the otherwise unfavorable localization of the protein in association with poorly aligned meshworks of the cell edge.”

Line 467:               “Other fimbrins”: does this mean all fimbrins except human ones?

So far, the calcium-independent activity of fimbrins has been reported for Schizosaccharomyces pombe, Tetrahymena, and Arabidopsis thaliana. For greater clarity, we have rephrased this sentence: “The regulation by Ca2+ is not universal, as fimbrins from Schizosaccharomyces pombe, Arabidopsis thaliana , and Tetrahymena are reported to bundle actin filaments in a Ca2+-insensitive manner  ” (page 14, lines 530-532).

Lines 485/501:      “Highly conserved” – fascin-3 only 28% identity – does this fit?

Thank you for helping to resolve the ambiguity. While mentioning “highly conserved,” we meant across species. Therefore, we replaced the word “highly” with “evolutionary” (page 15, lines 551). On the other hand, by stating, “Fascin-3 is the most diverse isoform that shares only 28% identity and 43% similarity with the other two isoforms (which share 72% similarity)” (page 15, lines 568-570), we are comparing the overall sequence similarity among different isoforms of fascin. Moreover, in the following line, we also mention that “Despite this moderate similarity, the residues involved in stabilizing the core and putative actin-binding sites are conserved ” (page 15, lines 570-571).

Line 509:               In Fig. 2 the three actin-binding sites are not recognizable.

The three actin-binding sites in fascin are formed upon the folding of four tandem β-trefoil folds (as mentioned in page 15, lines 575-577. For further clarification, Figure 2 is modified (now Figure 3).  

Lines 513/514:      Why a higher affinity for binding than for bundling?

The reviewer brought up an important question, for which we do not know the answer as we did not write those manuscripts. One possible explanation to the discrepancy is that bundling should be characterized by an apparent Kd due to the complexity of interaction via two domains and due to poor accessibility of F-actin subunits in the bundle for binding the subsequent molecules of fascin. The discrepancy could also arise from the experiments being conducted by different groups. However, given that none of these were discussed in the original manuscripts, we consider it inappropriate to discuss these points in the review. Also, please notice that the numbers are of the same order. Therefore, we only updated the manuscript in the following way: “Fascin cooperatively binds (Kd ~150 nM; [196,207]) and bundles (Kd ~270 nM; [116]) actin filaments” (page 15, line 580).

Lines 522/523:      I don’t understand the meaning of two sites and the angles between “each filament”.

We have rephrased the sentence in the following way: “In bundles, actin filaments are shifted relative to each other by 2.7 nm (an axial rise per actin subunit) and are bound at an average angle of 61°, resulting in an ordered hexagonal structure [73,185,206].” (page 16, lines 588-591).

Line 524:               “Compact and dense” bundles: what does that mean and how is that compatible with varying angles?

For this comment, we have rephrased these sentences (see above).

Line 542:               Please explain “can regulate cell motility in an actin-independent manner”.

For clarity, we have rephrased this sentence as: “Moreover, fascin also regulates cell motility in an actin-independent manner via direct interactions with microtubules  [208]”(page 16, lines 611-613).

Line 571:               Ankyrin repeats: are these designated AR in Fig. 2 ?

Thanks. For clarity, the full form of AR-“Ankyrin repeats” is added in Figure3 legend (earlier Figure 2) (page 9, 226, page 17, lines 641-642, page 18, lines 698 and 702).

Line 584:               “and contains”: the retinal epithelium or the vomeronasal organ?

We believe it is clear from the context, as the retinal epithelium is already indicated to contain Espin1.  “Similarly, a mature retinal sensory epithelium contains espin-1, -3, and -4, while the vomeronasal organ sensory cells (vestigial in humans but present in many other mammals and in reptiles) are enriched in espin-2 and -3, and contain trace amounts of espin-1, i.e., representing the developmental isoform distribution of the cochlear epithelium.” (page 17, lines 653-657)

Line 591:               “two conserved actin-binding sites”: In Fig. 2 only one is indicated.

The exact location of these actin-binding sites in the actin-binding domain is unknown; therefore, we deferred to show them in Figure 3 (previously Figure 2). However, in Figure 3 we have shown only Actin binding-domain and not actin-binding sites. For clarity, we have rephrased this sentence in the text: “Espins are monomeric proteins with C-terminal actin binding domain (ABD) responsible for a potent actin-bundling activity, suggesting the presence of at least two actin-binding sites within ABD [232].” (page 17, lines 664-666).

Line 649:               Similar to fascin-actin bundles … are uniformly oriented. In Table 1 and otherwise in the text, fascin results in mixed orientation.

Fascin stabilizes - and is primarily associated - with uniformly oriented filaments. We have made this statement throughout the manuscript. However, fascin is also found in bundles with mixed orientations of filaments, where it likely stabilizes a subset of parallel filaments. Table 1 and the text was updated to reflect this observation: “A likely role of fascin and espin in such bundles is the stabilization of a subset of filaments with uniform polarities; fimbrin/plastin, despite their compact size, can directly stabilize anti-parallel actin assemblies” (page 5, lines 146-148). Moreover, we have removed the sentence “Similar to fascin-actin bundles (page, 18, line 724).

Line 715:               The Tilney reference is missing.

The reference is added (page 20, line 797).

Line 753:               I do not understand how the elongation step smooths the (head-tail?) links.

As also suggested by the editor, we have rephrased these sentences and added new information for text clarity (page 21, lines 833-839) as follows: “Thus, during the initial stage of bristle formation in Drosophila, the nascent actin bundles form discrete discontinuous modules separated by gaps. In the later stages, a continued elongation links these short bundles to each other in a head-to-tail fashion, resulting in overlapping “graft” regions at the junction. These “grafts” regions are subsequently filled by actin filament elongation, thereby merging the adjacent short bundles into smooth and continuous actin bundles [270].”

Line 775:               What does here “Additionally” mean?

For clarity, we have rephrased this sentence and omitted the word “additionally”: “Actin bundles are directly or indirectly associated with transmembrane receptors (Eps8, TRPC, etc.; [279,281]) or transmembrane proteins (cadherin-23 and protocad-herin-15; [283]) at the tips of cell structures, such as microvilli, stereocilia, and filopodia. These proteins help in transducing mechanochemical signals and regulate actin bundle growth” (page 21, lines 860-865).

Line 815:               “does not support”. In other contexts, the cooperation of different bundles is emphasized.

The reviewer is correct, as the actin cytoskeleton is sufficiently complex to allow both types of behavior. As stated in the first line of this paragraph, “actin-bundling proteins spatiotemporally regulate the structure of bundles by cooperating with some and competing with other actin-bundling proteins (loose bundles with α-actinin and dense with fascin, fimbrin, and espin) [116,293]” (page 22, lines 893-895). Therefore, this sentence is not “out of focus” or “does not support” what we have previously mentioned (about the cooperativity of different bundling proteins). To avoid confusion we have rephared the sentence as follows “The segregated binding of bundling proteins occurs because they alter the topology of actin filaments proteins and their arrangement in the bundle. For instance, fascin binding over-twists the filaments in bundles, limiting the number of α-actinin binding sites [116,296]” (page 22, lines 899-903). In the same paragraph we also discussed the presence of different bundling proteins in specific actin structures, but their “localizations” are different: “α-actinin is restricted in cells to the basal portion of filopodia, while fascin and plastins are present along its length…….”(page 22, lines 896-897).

Line 870:               “fixed rate” – however line 875  3 sec-1 which is slower …

We thank the reviewer for catching this inconsistency. The first rate reflects the rate of pointed end depolymerization in vitro, which can be accelerated in vivo. We have modified the paragraph to address the uncertainty:

Actin treadmilling is the key element of actin filaments dynamic behavior. During treadmilling in vitro, ADP-G-actin is released from the pointed ends of aged filaments at a fixed rate of 0.25 sec-1 [313,314], converted to ATP-G-actin, and reused at the barbed ends. In a test tube, treadmilling maintains the overall filament length; in cells, global treadmilling enables constant re-organization of individual actin filaments and larger assemblies in response to cellular needs. In vivo, the treadmilling rate is not limited by the pointed-end depolymerization rate, as it can be dramatically accelerated by actin-binding proteins.” (page 23, lines 955-963).

Line 893:               “filament ends”: both ends?

Yes, as specified later in the text (page 24, lines 982-983).

Line 905:               Please specify “weak”: low affinity or slow?

In this case, the “weak” means both lower affinity and slower or sporadic severing, so we hope that the specification is unnecessary.

Line 906:               Please specify “cooperative”.

We would like to note that the “cooperative” terms is generally used for describing cofilin binding properties. We want to keep it as it is. But for clarification, by “cooperative binding” we meant that binding of cofilin to actin filaments is enhanced by the preceding binding event of cofilin, i.e. once cofilin is bound to actin filaments, binding of other cofilin molecule is favored.

Line 909:               “higher protein levels stabilize”, but line 932 “At high concentration, cofilin, promotes the disassembly…”

We agree and appreciate that this topic can be confusing as it reflects both in vitro and in vivo observations, which are often not easy to consolidate. We have mentioned explicitly that high/saturating cofilin concentration stabilizes actin filaments. On the other hand, at high concentrations cofilin rapidly disassembles actin-fascin bundles. The Reviewer may be confused here by the difference between actin filaments and actin bundles. To our knowledge both statements are correct. The explanation for high cofilin concentration, causing the disassembly of facsin bundles actin is mentioned (page 24, lines 1017-1018). Moreover to avoid confusion the text: “ Similarly to the effects of cofilin’s on single filaments, the molar  ratio of cofilin to actin in the cells determines cofilin’s effect on the bundles. At high concentrations, cofilin, promotes the disassembly of fascin-actin bundles, while their partial decoration by cofilin induces severing” is now omitted (page 24, lines 1021-1024).

Line 970:               “ADP-Pi ; actin”: why not ATP-actin in new filaments?

On actin filaments, newly added actin monomers immediately (within seconds) hydrolyze bound ATP to ADP-Pi, converting to ADP-Pi F-actin. As the Pi is released at a much slower rate (0.005 s−1), while ATP hydrolyses immediately, the majority of actin filaments barbed ends are in the ADP-Pi bound and not ATP bound state (PMID: 3566755, 3801442). Therefore, we wrote ADP-Pi- actin. This does not mean that ATP-actin cannot be dissembled.

Lines 1045/1046:  I don’t understand the meaning.

We have restated this part of the paragraph: “Despite the large amount of information available on this topic, the quantitative understanding of actin bundles formation is yet to be achieved. While the proteins involved in development of physiological bundles have been mainly identified due to the advances in mass spectrometry and related techniques, the time and sequence of their appearance in the bundles and their precise functions, are yet to be clarified.” (page 27, lines 1142-1147)

Reviewer 2 Report

The review-article of Sudeepa Rajan, Dmitri Kudryashov and Emil Reisler summarizes and carefully analyzes a huge and growing body of literature data on the actin bundles which are essential for many cellular functions. Generally, this review-article is comprehensive as it covers different aspects in the field of actin bundles formation, their dynamics and the role in the functioning of cytoskeleton, with citation of numerous works from the literature (363 references). The main aim of the review was to integrate and summarize current biochemical and structural information on several major actin-bundling proteins, namely α-actinin, fimbrin/plastins, fascin, and espin. The properties of these proteins, such as their domain organization, binding to F-actin, actin-bundling mechanism based on ability to cross-link actin filaments, isoforms of these proteins, etc. are described and analyzed in detail. Another part of the article (chapter 4) analyzes the main steps of bundle assembly (initiation, elongation, cross-linking by actin- bundling proteins), as well as disassembly/severing of F-actin and its bundles. In this part, the properties of actin-binding proteins involved in filament assembly and disassembly (such as ADF/cofilin family proteins, capping proteins (CP or CapZ), gelsolin, profilin, different types of myosins, etc.) are also briefly discussed, along with their role in these processes. The manuscript is well written, good-organized, and illustrated by 3 good-quality figures. In my opinion, this review-article, which summarizes and carefully analyzes recent literature data on the proteins involved in actin bundles assembly and disassembly, will be very useful for all scientists working in the field of actin functions in the cells, and therefore it certainly can be published in Biomolecules (in the Special Issue "Actin and Its Associates: Biophysical Aspects in Functional Roles") after few small corrections according to the following remarks.

 Minor remarks

 1). Page 6, line 191: to replace “3.1α-. Actinin” by “3.1. α-Actinin”.

 2). Page 7, Figure 2: It seems to me, it would be better to place this figure that shows domain organization of various actin-bundling proteins before part 3.1 devoted to α-Actinin, but not inside this part.

 3). The title of Figure 2 on page 7 ("Domain organization of actin-bundling protein") should be replaced by "Domain organization of actin-bundling proteins" as various proteins which differ in their domain organization are shown in this figure. 

Author Response

We thank the reviewer for the valuable and positive comments on our manuscript. These comments have helped us enhance our manuscript’s quality and readability. The issues pointed out to us are addressed and answered below. Kindly use tracked .pdf version of the manuscript for referring to page and line numbers.

 Minor remarks

 1). Page 6, line 191: to replace “3.1α-. Actinin” by “3.1. α-Actinin”.

We thank the reviewer for noticing this typo. It has been corrected (page 9, lines 247).

 2). Page 7, Figure 2: It seems to me, it would be better to place this figure that shows domain organization of various actin-bundling proteins before part 3.1 devoted to α-Actinin, but not inside this part.

We have changed the position of Figure 2 (now Figure 3) as suggested by the Reviewer.

 3). The title of Figure 2 on page 7 (“Domain organization of actin-bundling protein”) should be replaced by “Domain organization of actin-bundling proteins” as various proteins which differ in their domain organization are shown in this figure. 

Corrected (now Figure 3).

Reviewer 3 Report

The review “Actin bundles dynamics and architecture” submitted by Sudeepa Rajan, Dmitri S. Kudryashov and Emil Reisler is a thorough  analysis of F-actin bundle formation and detailed description of four protein  families involved in bundling. I think the review is of great interest for researchers studying cytoskeleton, and I highly recommend it for publication.  

I have only few minor comments:

Line 29 “…filaments (F-actin), also known as microfilaments or thin filaments” and line 148 “…regions that contain thin (actin) filaments…” Different from microfilaments that are considered to be same as actin filaments, thin filaments in modern cell biology are not same as actin filaments but are defined as filaments formed by actin, tropomyosin and troponin. These sentences should be re-written to make clear that thin filaments and actin filaments are not synonyms.

Line 188 “…In addition to actin-binding domains, most bundling proteins contain other functional domains (such as the LIM, ankyrin, WH2, EF-hand domains, spectrin repeats, etc.)” WH2 is an actin-binding domain but this sentence sounds like it is not.

Line 321 “…and actin's limited proteolysis by calpain-1…” Shouldn’t it be actinin instead of actin?

Legend to Figure 2 should have explanation that AR stands for ankyrin repeats and Pro is for Pro-rich regions.

Lines 277, 321, 447, 529-533. Sometimes the same isoform may have different number of residues in different species, therefore it would be helpful to readers if the authors add species information for proteins with mutated or modified residues mentioned in the review.

Lines 168, 285. Font sizes are different.

Author Response

Reviewer 3

We thank the reviewer for the positive comments on our manuscript. These comments have helped us enhance our manuscript’s quality and readability. The issues pointed out to us are addressed and answered below. Kindly use tracked .pdf version of the manuscript for referring to page and line numbers.

I have only few minor comments:

  1. Line 29 “…filaments (F-actin), also known as microfilaments or thin filaments” and line 148 “…regions that contain thin (actin) filaments…” Different from microfilaments that are considered to be same as actin filaments, thin filaments in modern cell biology are not same as actin filaments but are defined as filaments formed by actin, tropomyosin and troponin. These sentences should be re-written to make clear that thin filaments and actin filaments are not synonyms.

We thank the reviewer for pointing out to this difference. To address the concern, we removed the term “thin filaments” from the sentence (page 1, line 29). The text is rephrased now to “Each sarcomere contains thick (myosin) filaments in the center and is flanked by regions that contain thin filaments (actin filaments decorated by tropomyosin and troponin).” on page 6, line 181.

Line 188 "…In addition to actin-binding domains, most bundling proteins contain other functional domains (such as the LIM, ankyrin, WH2, EF-hand domains, spectrin repeats, etc.)" WH2 is an actin-binding domain but this sentence sounds like it is not.

The reviewer is again very observant. For clarity, we have rephrased this sentence: “In addition to F-actin-binding/bundling domains, most bundling proteins contain other functional domains (such as the LIM, ankyrin, WH2, EF-hand, spectrin repeats, etc.” (page 9, line 242).

Line 321 “…and actin's limited proteolysis by calpain-1…” Shouldn't it be actinin instead of actin?

Yes, it should be actinin. Thank you for bringing this to our attention (page 12, line 374).

Legend to Figure 2 should have explanation that AR stands for ankyrin repeats and Pro is for Pro-rich regions.

 As noted also by reviewer 1, the full form of AR and Pro has been now added in the Figure 3 legend (previously Figure 2). Additionally, the AR full form is also mentioned in the text on page 17, lines 641-642 and page 18, line 698.

Lines 277, 321, 447, 529-533. Sometimes the same isoform may have different number of residues in different species, therefore, it would be helpful to readers if the authors add species information for proteins with mutated or modified residues mentioned in the review.

Now added as following : page 11, line 373; page 12, line 374, ; page 14, lines 510, 523; page 16, lines  598, 602.

Lines 168, 285. Font sizes are different.

Corrected.

Round 2

Reviewer 1 Report

Table 1                  What does “70; 250-300 µm” mean?

Lines 102:        Delete “Like filopodia” because they don’t arise at the ventral surface.

Line 105:               Podosomes play critical roles in phagocytosis. Does that mean that phagocytosis requires podosomes?

Fig. 1A:                 Are capping proteins and myosin I really disassembly factors? The legend says that capping proteins stabilize filament ends.

Fig. 1B:                 The new images are o.k. although they do not show the details I hoped to see. Legend: add space between “0.05µm”.

Fig. 2:                    Is actin really bundled in sarcomeres?
The line “Actin bundles in migratory cell” should be moved to clearly indicate the corresponding cell.

Line 193:           The cortex of Amoeba proteus has a thick layer of actin, but I would not say that this is general. Please specify by giving measure of thickness.

Line 275:               delete “the”

Line 311:               Suggestion: “Similar as in other tandem CH domain proteins (…”

Line 345-349:       I do not understand the connection of low affinity and the subunits in a long-pitch turn. Isn’t that a matter of the distance between ABD’s in α-actinin?

Line 352:               Not clear where “define” refers to. I had hoped to see an image of distorted diamond-shaped lattices.

Line 444:               “therefore”. Probably “accordingly” is more appropriate .

Line 497:               add space to “12µm”.

Line 539-540:   Not clear to me how severing improved bundling in TIRF experiments.

Line 668-672:     I do not understand “which correlates with higher espin. levels. And also not the following statements on elongation.

Line 837:           When the bundles are already linked to each other, what does it mean that grafts are subsequently filled?

Line 900:               is “because” the right term?

Line 962/964:       Treadmilling can be accelerated by actin-binding proteins but it is slower in

Line 977:               How do formins increase the G-actin concentration?

Author Response

We have addressed the comments point by point.
